# Measurements and modelling of surface-atmosphere exchange of microorganisms in Mediterranean grassland

Carotenuto Federico[1,2], Georgiadis Teodoro[2], Gioli Beniamino[2], Leyronas Christel[3], Morris Cindy E.[3], Nardino Marianna[2], Wohlfahrt Georg[1], Miglietta Franco[2,4,5]

[1]: Institute of Ecology, University of Innsbruck, Sternwartestrasse 15, Innsbruck, A-6020, Austria

[2]: Institute of Biometeorology (IBIMET), Consiglio Nazionale delle Ricerche (CNR), Via G. Caproni 8, I-50145, Firenze (Italy)

[3]: Plant Pathology Research Unit, French National Institute for Agricultural Research (INRA), Allée des Chênes 67, Montfavet, F-84143, France

[4]: FoxLab, Joint Research Unit Fondazione Edmund Mach - CNR IBIMET, Via E. Mach 1, San Michele all'Adige, I-38010, Italy

[5]: IMèRA, Universitè Aix-Marseille 2, Place le Verrier, Marseille, F-13004, France.

*Correspondence to*: Carotenuto Federico (f.carotenuto@ibimet.cnr.it)

**Abstract.** Microbial aerosols (mainly composed by bacterial and fungal cells), may constitute up to 74 % of the total aerosol volume. These biological aerosols are relevant not only from the point of view of the dispersion of pathogens, but also has geochemical implications. Some bacteria and fungi may, in fact, serve as cloud condensation or ice nuclei, potentially affecting cloud formation and precipitation and are active at higher temperatures compared to their inorganic counterparts. Simulations of the impact of microbial aerosols on climate are still hindered by the lack of information regarding their emissions from ground sources. This present work tackles this knowledge gap by i) applying a rigorous micrometeorological approach to the estimation of microbial net fluxes above a Mediterranean grassland and ii) developing a deterministic model (the PLAnET model) to estimate these emissions on the basis of a few meteorological parameters that are easy to obtain. The grassland is characterized by an abundance of positive net microbial fluxes and the model proves to be a promising tool capable of capturing the day-to-day variability in microbial fluxes with a relatively small bias and sufficient accuracy. PLAnET is still in its infancy and will benefit from future campaigns extending the available training dataset as well as the inclusion of ever more complex and critical phenomena triggering the emission of microbial aerosol (such as rainfall). The model itself is also adaptable as an emission module for dispersion and chemical transport models, allowing further exploration of the impact of land-cover driven microbial aerosols on the atmosphere and climate.

## 1 Introduction

Vegetated land surfaces, and plant leaves in particular, harbor a large number of microorganisms that can be transported by wind. It has been estimated that the planetary phyllosphere harbors about $10^{24}$ to $10^{26}$ bacterial cells (Morris et al., 2002) of the $10^{30}$ that live on Earth (Whitman et al., 1998). Up to $10^7$ bacteria per $cm^2$ are present on leaf surfaces (Morris et al., 2004), and plant materials are considered the largest source of fungal spores in the atmosphere (Burge, 2002). All of these organisms can be transported into the atmosphere by wind (Delort et al., 2010), as was shown experimentally in an artificial wind-gust chamber (Lighthart et al., 1993). Atmospheric transport can involve both multiple short-distance events (Brown and Hovmøller, 2002), as well as single long-range movements. The latter are well known to transport desert dust (Rosselli et al., 2015;Peter et al., 2014;Kellogg and Griffin, 2006;Griffin, 2007;Weil et al., 2017), while long-range transport of epiphytic organisms living in plant canopies is much less documented. Nevertheless living and dead microorganisms are part of primary biological aerosols (PBAs) that contribute 13 to 74 % of the entire aerosol volume globally (Graham et al., 2003). Furthermore, microorganisms can be found in cloud water droplets. Water sampled from clouds over alpine regions in France and Austria contained about $2 \times 10^4$ $mL^{-1}$ of bacteria (Amato et al., 2007;Bauer et al., 2003), while fungi were at least an order of magnitude lower. Different bacterial species were also found in fog droplets of the Po plain in Italy (Fuzzi et al., 1997), as well as in clouds over Scotland (Ahern et al., 2007).

The presence of microorganisms in the atmosphere may be relevant to climate processes given that some of these microorganisms can serve as cloud condensation or ice nuclei (Möhler et al., 2007;Morris et al., 2004;Szyrmer and Zawadzki, 1997;Hoose et al., 2010), potentially affecting cloud formation and climate (Amato et al., 2007). Some microbial species, in fact, are able to freeze water at temperatures significantly warmer than those induced by non-biological ice nucleators (-2 to -7 °C versus <-10 °C or -15 °C) (Morris et al., 2004). In the past, only few attempts have been made to directly measure the flux of bacteria from plant canopies (Lindemann et al., 1982;Lindemann and Upper, 1985;Lighthart and Shaffer, 1994;Crawford et al., 2014). Direct eddy covariance measurements of aerosols exchange in tropical forests, where PBAs represent a significant fraction of the airborne particulate matter (Graham et al., 2003), were also performed by Ahlm et al. (2010) and Whitehead et al. (2010), potentially giving a proxy for microbial emission in tropical ecosystems. The mass of PBAs that is actually released by different land use types under different conditions and, more importantly, the specific composition of such fluxes and their quantification remains so far mostly unresolved. As a consequence, numerical quantification of microbial emissions, as well as investigations of the effects of living biological particles on the atmosphere and the water cycle have been limited to highly idealized scenarios. Lighthart and Kirilenko (1998) attempted to simulate summertime diurnal emission dynamics, but in their work net upward fluxes were a function of time and solar radiation-dependent microbial death only. Population dynamics in the phyllosphere and atmospheric turbulence were not accounted for. In their attempt to simulate global impacts of microbial particles on the water cycle, Hoose et al. (2010) and Sesartic et al. (2012) used fixed values of bacterial emission fluxes from different ecosystems, while Heald and Spracklen (2009) used

mannitol as a proxy to evaluate fungal contributions to PBAs. Burrows et al. (2009a) modeled global emissions using data about airborne concentrations of microbes reported in the literature and derived fluxes from such proxies.

The aim of this paper is twofold: i) to increase knowledge of microbial emissions by quantifying fluxes by means of a more complex micrometeorological method compared to earlier attempts and ii) to propose a deterministic model to estimate both

on-ground population dynamics and the associated atmospheric exchange processes.

The latter model was calibrated with microbial flux measurements made episodically over 3 years (between 2008 and 2010), while a second measurement campaign (2015) was used to validate its performance.

## 2 Materials and Methods

### 2.1 Flux Measurements

Microbial fluxes were measured during two field campaigns in a pasture in Montfavet, France, (43.95° N, 4.88° E, 32 m a.s.l.). Measurements were made between 7 and 11 July 2015 and again between 26 September and 1 October 2015. The pasture was a typical Mediterranean grassland dominated by grasses in an area surrounded by similar land uses and with no significant orographic features. The vegetation status was different during the two campaigns: in July the grassland showed visible signs of water stress, but no more than 20 % of the leaves were chlorotic and dry. In September the grassland was instead well

developed and mostly green. The mean height of the canopy was approximately 20 cm for both campaigns. The field was mainly covered with clover (*Trifolium spp.*) and ryegrass (*Lolium perenne*), was not intensively grazed and was not actively managed during the measurement campaigns with no mowing or irrigation. Profiles of wind speed, air temperature and viable aerosols were made at two heights (≈67 and ≈227 cm), while a sonic anemometer (USA-1, Metek, Elmshorn, DE; located at ≈300 cm) measured 3D wind components and the sonic temperature

at 20 Hz frequency. In September, an open-path infra-red gas analyzer (Li-7500, LiCor, Lincoln, Nebraska, USA) along with a differential infra-red gas analyzer (Li-7000, LiCor, Lincoln, Nebraska, USA) were added to the set-up in order to measure the $CO_2$ and $H_2O$ gas exchange concurrently by the eddy-covariance and the flux-gradient method (Baldocchi et al., 1988). This setup (Fig. 1) allowed to assess the performance of the flux-gradient method of estimating, water vapor fluxes vs. the respective fluxes directly measured by the eddy covariance method. Viable bioaerosols were sampled with Burkard jet

samplers (Burkard Manufacturing Co. Ltd., Rickmansworth, UK). The samplers operated at a flow rate of 500 l min$^{-1}$ and particles were collected on Petri dishes containing 10 % tryptic soja agar (1.7 g tryptone, 0.3 g peptone soja, 0.25 g glucose, 0.5 g NaCl, 0.25 g $K_2HPO_4$, 15 g agar per L). After sampling, the dishes were incubated at 25 °C and microbial colonies were counted after 24 h of incubation and for up to 3 days. Such medium was non-selective allowing the growth of both bacteria and fungi.

Sampling with each Petri dish lasted for 14 minutes and for every day a handling blank was incubated alongside the sampled plates.

The design of the virtual impactor followed good design practices with a direct alignment of the nozzle and the collection probe (i.e. the still air chamber) and diameter of the collection probe (0.08 m) at least 40 % larger than the nozzle diameter (Marple and Olson, 2011). Data from literature indicate a sampling efficiency ranging from 80 to 100 % for mildew spores (Schwarzbach, 1979). Given the Burkard sampler's high flowrate, sampling happens at a super-iso-mean-velocity compared with external wind speed. The sampling efficiency is therefore expected to decrease for larger particles proportionally with the ratio between external wind speed and the Burkard's sampling speed (Brockmann, 2011).

A series of 10-minute samplings with the Burkard samplers kept at the same height was done to evaluate the minimum resolvable gradient (MRG) following Eq. (1) below (Edwards et al., 2005;Fritsche et al., 2008). An ANOVA was used to verify the absence of significant difference in the number of colonies counted between the two samplers.

$$MRG = (|(\overline{A - B})|) + \sigma(A - B) , \qquad (1)$$

where $A$ indicates the sequence of number of colonies in the top sampler, $B$ in the bottom sampler and $\sigma$ the standard deviation of the respective differences.

The flux-gradient method was used to estimate microbial fluxes from concentrations measured with the Burkard samplers. This methodology has been widely used to measure atmospheric fluxes of different scalars such as hydrogen (Meredith et al., 2014), nitrates and nitrogen compounds (Beine et al., 2003;Griffith and Galle, 2000;Taylor et al., 1999), mercury (Edwards et al., 2005;Fritsche et al., 2008;Lindberg et al., 1995) and particulate matter (Bonifacio et al., 2013;Kjelgaard et al., 2004;Park et al., 2011;Sow et al., 2009). The method follows the Monin-Obukhov similarity theory (Monin and Obukhov, 1954) and, therefore, assumes that in the atmospheric surface layer the flux of a certain scalar is a function of the gradient of the scalar measured at different heights, the heights themselves ($z_i$), and a transport velocity which is dependent on atmospheric turbulence and stability (a more detailed description of the methodology is provided in the supplementary material).

The measurement setup was chosen to avoid sampling in the roughness sub-layer, where the scaling principles do not hold and a conservative roughness length ($z0 = 0.15$ m) was chosen (Businger, 1986). This length was adequate to obtain a $z/z0 > 1$ (Businger, 1986) at both the low and high sampling heights, thus offsetting the presence of upwind obstacles that were within the range of the required horizontal surface uniformity ($\approx 25$ times z meters, Irvine et al. (1997)). Coherently with the cited literature, fluxes are reported from the perspective of the atmosphere as positive when upward (i.e.: emissions) and negative when downward (i.e.: sinks).

A similar setup was employed between 2008 and 2010 to measure PBAs in an area very close by (43.91° N, 4.87° E and roughly 30 m a.s.l.). The Bukard samplers were deployed in a gradient configuration (at 50 and 250 cm above ground) along with a nearby sonic-anemometer stationed at roughly 230 cm above ground. No trace gases measurements were made during this period. The experimental field for these previous campaigns was covered with herbaceous species with similar habitus such as cocksfoot (*Dactylis glomerata*), ryegrass, tall fescue (*Festuca arundinacea*) and alfalfa (*Medicago sativa*). In the 2008-2010 campaigns a different methodology was used to assess the sampling differences between the two Burkard samplers. The

two samplers were put together and a serial dilution of *P. syringae* was aerosolized. Three replicate samples were taken per each dilution ($10^2, 10^3$ and $10^4$ live bacteria per ml) and no statistical differences were detected in the CFUs sampled by the Burkard samplers, with the single exception of one replicate at the $10^3$ dilution. All the tests were conducted with an open petri dish used to verify the deposition of the aerosolized spray.

## 2.2 The Plant-Atmosphere Epiphytic Transport (PLAnET) model

The model estimates microbial fluxes from the phyllosphere via a set of meteorological variables (air temperature, friction velocity and wind speed), the leaf area index (LAI) and atmospheric pressure. The model assumes that soil is an insignificant source of microorganisms for the atmosphere compared to the plant canopy. Other studies have considered that plant materials

are the largest source of fungal spores in the atmosphere (Burge, 2002) and have shown that bacterial fluxes are higher over plants, except in cases of relatively rare events such as dust storms (Lindemann et al., 1982;Lindemann and Upper, 1985). This is in agreement with the finding that higher wind speeds are necessary to free a particle from soil rather than from the plant canopy (Jones and Harrison, 2004).

The model is based on three fundamental modules:

1. **Source**: Microbial population dynamics are driven by temperature, humidity, immigration/emigration phenomena, competition (both between the microbial species and between plants and pathogens) and these factors change throughout space and time. To reduce the complexity of such interactions, the PLAnET model limits the representation of the microbial source to its main driver as a temperature-dependent growth function.

2. **Removal**: this is an energy driven processes. Wind shear and buoyancy act on the microbial population making a fraction

of it airborne.

3. **Deposition:** Microbial deposition is computed as the product of a settling velocity and an airborne concentration estimated on LAI. The settling velocity itself is a linear combination of gravitational settling (computed following Kulkarni et al. (2011)) and impaction/interception (computed following Slinn (1982)).

The gross upward flux of microbes into the atmosphere was simulated following a logistic equation (Eq. (2)), assuming the

existence of a threshold friction velocity (Aylor et al., 1981;Geagea et al., 1997). When simulating dust emissions it is generally assumed that there is a linear (Raupach and Lu, 2004) or exponential (Gillette and Passi, 1988) relationship between upward dust flux and friction velocity ($u_*$), due to the existence of saltation bombardment (Raupach and Lu, 2004;Dupont et al., 2013). Phyllosphere microbial populations are far from being comparable to the soil surface on which such bombardment occurs and, therefore, a different mechanism has been chosen in the present context. It is assumed that no saltation mechanisms can

intervene in amplifying particle removal and, therefore, the upward flux will saturate at a certain $u_*$. The idea for this representation of the upward flux is summed up in Fig. 2.

$$F_e = \left\{ \left[ m_1 \exp^{\left(-m_2 \exp^{(-m_3 u_*)}\right)} \right] \right\} \frac{N}{k_{max}}, \tag{2}$$

In Eq. (2) $F_e$ is the gross upward flux (in CFU m$^{-2}$ s$^{-1}$); $m1$, $m2$ and $m3$ are respectively 30 CFU m$^{-2}$ s$^{-1}$, 256.26 CFU m$^{-2}$ s$^{-1}$ and 19 and were derived though a curve fitting to the Lighthart and Shaffer (1994) flux data with FOOTPRINT92 $u_*$ data (Lighthart and Shaffer, 1994) and the model calibration procedure; $N$ is the phyllosphere population in the model (in CFU m$^{-2}$), and $k_{max}$ is the maximum allowed microbial population ("carrying capacity" in CFU m$^{-2}$).

If only wind speed, instead of wind speed and friction velocity, is provided as an input, the model calculates $u_*$ using Eq. (3):

$$u_* = k\, u \ln\left(\frac{z}{z0}\right), \tag{3}$$

In Eq. (3) $k$=0.4 and is the Von Karman constant; $u$ is the wind speed (in m s$^{-1}$); $z$ the sampling height in meters and $z0$ the roughness length (=0.15 m).

The gross downward flux (i.e.: deposition) is modeled following Eq. (4)

$$F_d = \left(V_g + V_i\right)C_a , \tag{4}$$

$F_d$ is the deposition flux (CFU m$^{-2}$ s$^{-1}$), $V_g$ the gravitational settling velocity (m s$^{-1}$), $V_i$ (m s$^{-1}$) the settling velocity due to impaction/interception from roughness elements and $C_a$ the airborne concentrations of microorganisms (CFU m$^{-3}$).

$V_g$ is calculated following (Kulkarni et al., 2011) (Eq. 5):

$$V_g = \frac{g \rho_p d^2 C_c}{18\eta} , \tag{5}$$

where $g$ is the gravitational acceleration (9.81 m s$^{-2}$); $\rho_p$ is the particle density (1100 Kg m$^{-3}$, Cox and Wathes (1995)); $d$ is the particle diameter (3.3 x10$^{-6}$ m, Raisi et al. (2013) and Schlesinger et al. (2006)); $C_c$ is the Cunningham slip correction factor; and $\eta$ is the air viscosity (1.83 x10$^{-5}$ Pa s, Kulkarni et al. (2011)).

The term $V_i$ represents the effect of interception/impaction on particle deposition and it has been computed following Slinn (1982)

$$V_i = Cd\, u_r \left(1 + \frac{u_h}{u_r}\frac{1-\epsilon}{\epsilon+\sqrt{\epsilon}\tanh\gamma\sqrt{\epsilon}}\right)^{-1} \tag{6}$$

where $C_d$ is the ratio between $u_*^2$ and $u_r^2$; $u_r$ represents wind speed measured at a reference height (in m s$^{-1}$); $u_h$ wind speed measured at canopy height (in m s$^{-1}$) and $\epsilon$ the particle/canopy-element collection efficiency (adimensional). The latter has been computed following Slinn (1982), but without accounting for diffusional effects ($E_B$ in the cited paper) since they are not significant for particles >1 μm (Wiman and Ågren, 1985). Eq. (6) has the form of a velocity (being essentially a scaling factor for wind speed) and, when combined with $V_g$ (see Eq. (4) and (5)), determines the actual particle deposition velocity. To solve Eq.(6) two wind speeds are needed (measured at canopy height and at a reference height above canopy), whose ratio can be expressed as:

$$\frac{u_h}{u_r} = \frac{u_*}{ku_r}\, ln\frac{l}{z_0} \tag{7}$$

where $l$ is a characteristic eddy size in the canopy (expressed in m), which, in the present simplified implementation of the

Slinn model, has been considered equal to canopy height (h, Slinn (1982)). Following Slinn's work, the parameter $\gamma$ has been assumed to equal h$^{1/2}$, while the other constants were set as: $c_v/c_d$ = 1/3, Ă=10 μm, Â=1 mm, $F$=1%, b=2, $c_{Stk}$=1.

The term $C_a$ has been calculated as a characteristic seasonal airborne concentration for a Mediterranean grassland via a linear relationship between LAI values and average concentrations between the top and bottom sampler during the 2008-2010 campaign following Eq. (8):

$$C_a = p_1 LAI + p2 \ , \tag{8}$$

In Eq. (8) $p1$=26.99 CFU m$^{-3}$ and $p2$=115.9 CFU m$^{-3}$. LAI values used in this study were obtained from MODIS data (Myneni et al., 2015). Four 500 m pixels were averaged in space and interpolated in time to the half hourly time series from the 4-day LAI time-step of the satellite data. The average between-pixel standard deviation was quite consistent, varying slightly between $\pm$ 0.32 and $\pm$ 0.35 across all the simulated years.

The actual net PBAPs flux at a given time ($F_n$, CFU m$^{-2}$ s$^{-1}$) is computed following Eq. (9):

$$F_n = F_e - F_d, \tag{9}$$

The phyllosphere microbial population ($N$, see Eq. (2)) is modeled following Eq. (10):

$$N = rN - (F_n \xi) \tag{10}$$

The growth rate $r$ of Eq. (10) is modeled as a temperature-driven process in Eq. (11) (Yan and Hunt, 1999;Yin et al., 1995;Magarey et al., 2005):

$$r = \begin{cases} 0 & if \ T < T_{MIN} \\ \left(\frac{T_{MAX}-T}{T_{MAX}-T_{OPT}}\right)\left(\frac{T-T_{MIN}}{T_{OPT}-T_{MIN}}\right)^{\left(\frac{T_{OPT}-T_{MIN}}{T_{MAX}-T_{OPT}}\right)} c & if \ T_{MIN} \leq T \leq T_{MAX} \\ 0 & if \ T > T_{MAX} \end{cases} , \tag{11}$$

In Eq. (9) $T_{MIN}$ ,$T_{MAX}$ and $T_{OPT}$ are, respectively, the minimum, maximum and optimal growth temperatures (in °C). $c$ is a calibration constant accounting for the unknown doubling time of the microbes in the phyllosphere. For the purpose of Eq. (10) the net flux is multiplied by the model time step ($\xi$=1800 s) making the units of the second right hand term coherent with the units of the first right hand term ($rN$, CFU m$^{-2}$).

The model also includes two thresholds: a minimum ($k_{min}$, a number of microorganisms sheltered by wind action following the concept of Waggoner (1973)) and a maximum population size ($k_{max}$ or "carrying capacity" which is the maximum population that an ecosystem can sustain indefinitely, Verhulst (1838)). When the population falls below $k_{min}$ no removal can happen and if the population overshoots $k_{max}$, no growth can happen. Since the model is focused on phyllosphere dynamics, $k_{max}$ is appropriately scaled with LAI in order to represent plant senescence and, therefore, the reduced availability of space and resources. The model starts from an estimate of the initial population ($N_0$), representing the "boundary condition" for the modeled processes at the start of the simulation and proceeds with half-hourly time-steps until the end of the simulation period.

## 2.3 Model Calibration and Sensitivity Analysis

The model was run between 1 January 2008 and the 31 December 2010, assuming that the microbial population in the phyllosphere at the beginning of the period was equal to $k_{min}$ due to low LAI and temperature. The error metric to evaluate model performance is described by Eq. (12):

$$\varepsilon = |(1 - |s|)| + |o| + |(1 - |r^2|)|, \tag{12}$$

In Eq. (12), $\varepsilon$ represents the error metric, $s$ the slope of the linear relationship between measured and modeled net fluxes, $o$ the offset of this relationship and $r^2$ is the correlation coefficient. The function receiving the model parameters as input and returning $\varepsilon$ as an output was passed to MATLAB's *fmincon* interior-point algorithm (Byrd et al., 1999;Byrd et al., 2000;Waltz et al., 2006) as the objective function for minimization. The algorithm was run iteratively through MATLAB's *GlobalSearch* function in order to avoid finding a set of parameters satisfying only a local minimum. To avoid mathematically sound, but non-realistic solutions, *GlobalSearch* was looking for minima only within a bounded parameter space. The upper and lower bounds of the parameter space are shown in Table 1. The percentage of leaf area exposed to turbulence was arbitrarily estimated to correspond to 5 % of the average leaf area density of the grassland that was set at 94 g m⁻² (Sims and Singh, 1978). The latter assumption was needed to scale the measurements in CFU g⁻¹ of Hirano and Upper (1986) and Wilson and Lindow (1994) to the units needed by the model (CFU m⁻²).

The calibrated model was then run on the data collected in 2015 in order to assess its performance on a dataset not used for training. Optimal temperature was not entered as a calibration parameter but was assumed to be halfway through between the $T_{MIN}$ and $T_{MAX}$ chosen by the optimization algorithm.

Sensitivity of the model was analyzed by computing new values of $\varepsilon$ by varying each parameter by plus and minus 10 %. For each parameter a mean $\varepsilon$ was computed by averaging the two errors resulting by the up and down modifications. Finally, a sensitivity metric was obtained by simply subtracting the $\varepsilon$ obtained by the optimization procedure from the average error of each parameter.

## 3 Results

### 3.1 Field Measurements

During the 2015 campaigns temperature ranged between 13.4 and 34.1 °C (mean 25.2 ± 6 °C, right y-axis Fig. 3a). Wind speed fluctuated between 0.2 and 5.3 m s⁻¹ (mean 2.1 ± 1.2 m s⁻¹, left y-axis Fig. 3b) with a general northerly wind direction (right y-axis Fig. 3b). During the same campaign fungal colonies dominated the microbial colonies growing on culture media, but bacterial-like colonies were also present. Measured microbial fluxes varied both between and within the days of the two field campaigns (July and September, left y-axis Fig. 3a) with individual flux measurements being above the MRG in 60.6 % of all cases. Unreliable fluxes were unevenly distributed between July and September and included all negative fluxes (i.e.,

deposition, left y-axis Fig. 3a). In 2015, the plant canopy was a net microbial emitter (left y-axis Fig. 3a), with net fluxes ranging between 0.2 and 28.5 CFU $m^{-2}s^{-1}$. An overview of the relationship between counted CFUs, MRG and estimated fluxes is presented in Fig 4. In September 2015, fluxes of water vapor directly measured by eddy-covariance were compared with the ones resulting from the application of the flux-gradient method, yielding a high correlation between the two ($r^2= 0.70$) and

with minimal bias (y = 1.05x – 0.08; RMSE = 0.79) (Fig. 5), thus showing the absence of divergences between the two methods. The measurements between 2008 and 2010 were made in different seasons, resulting in a wider range of temperatures spanning from 7.9 to 28.1 °C (mean 18.5 ± 4.8 °C, right y-axis Fig. 6a). Wind speed was consistent with the 2015 campaign, ranging from 0.4 to 5.8 m $s^{-1}$ (mean 2.4 ± 1.4 m $s^{-1}$, left y-axis Fig. 6b) with a mainly northerly wind direction (with the exception of 2009 for which no wind direction data were available, right y-axis Fig. 6b). Microbial fluxes within these three years spanned

a wider range of magnitude, varying between -5.2 and 57.1 CFU $m^{-2}s^{-1}$ (left y-axis Fig. 6a). The average flux between 2008 and 2010 was close to the 2015 average (8.3 CFU $m^{-2}s^{-1}$ in 2008-2010 versus 10.6 CFU $m^{-2}s^{-1}$ in 2015), while the standard deviation was higher (11.1 CFU $m^{-2}s^{-1}$ in 2008-2010 versus 6.2 CFU $m^{-2}s^{-1}$ in 2015). Few negative fluxes were registered in 2008-2010, that represented only 16.8 % of the total, confirming that the sampling site tended to be a net microbial emitter, rather than a sink.

**3.2 Model Calibration**

The results of the optimization are resumed in Table 2 where the chosen parameters are reported along with the respective sensitivity value.

All the chosen values fell within the imposed boundaries (see Table 1). The optimization procedure was able to find a meaningful optimum as it is deducible by looking at the sensitivity values reported in Table 2. Any variation of a parameter

results in a worsening of the error metric (i.e.: a positive sensitivity value), even if the model is not equally sensitive to all the parameters. More specifically, the minimum temperature regulating the growth curve of the microorganisms is the one with the highest impact on model performance, while the minimum population size ($k_{min}$) seems to have the least impact. The latter result also suggests that the approximation made concerning the percent of leaf area exposed to turbulence (i.e. 5 %) is not critical.

Relationships between measured and modeled fluxes with an optimal set of parameters are reported in Fig. 7 for both the calibration set (2008-2010, Fig. 7a) and the validation campaigns (2015, Fig. 7b). The model is consistent between the two campaigns, explaining roughly 55-70 % of the variance ($r^2$ for 2008-2010 is 0.54 while $r^2$ for 2015 is 0.68) and it does it with a small offset (0.28 in 2008-2010 and -3.75 CFU $m^{-2}$ $s^{-1}$ for 2015). The model still has a bias in the flux estimation: it tends to underestimate the fluxes during the calibration campaign (slope of the regression of 0.70) and overestimate them during the

2015 field campaigns (slope of the regression 1.31). The model has an RMSE of 5.82 CFU $m^{-2}$ $s^{-1}$ in 2008-2010, while it is 2.78 for 2015.

Interestingly, while a clear dependence of the measured fluxes on atmospheric turbulence ($u_*$) was frequently observed, $u_*$ was not always correlated with flux contrary to what might be expected. On some occasions, the measured microbial fluxes were

much lower than predicted by Eq. (2) which directly scales the effect of turbulence on the microbial fluxes. This observation is consistent with the assumptions made in the PLAnET model, where the actual microbial flux is indeed driven by turbulence, but also constrained by the rate at which microorganisms multiply and by the size of the microbial population in the phyllosphere. This is represented graphically in Fig. 8a and 8b. When the population is close to the minimum population ($k_{min}$),

even very high turbulence (mean $u_*$ for Fig. 8a is 0.49 m s$^{-1}$) will not elicit significant upward net fluxes. Conversely, when the microbial population is large (in Fig. 8b above 4.8 x 10$^5$ CFU m$^{-2}$), even low turbulence (mean $u_*$ for Fig. 8b is 0.39 m s$^{-1}$) will generate significant upward net fluxes. Accordingly, the model was able to capture the variability in the amount of CFUs that can be instantaneously transported into the atmosphere, thus predicting complex interactions between weather conditions, microbial population densities and the actual flux. This latter result suggests that for organic particles the simple knowledge

of the transport field may not be enough: microbial populations have their own inherent dynamics (growth, death, immigration and emigration) that influence the amount of microbial cells available for transport. This phenomenon is clearly absent in the modelling of dispersion of inorganic dust particles.

Simulated daily sums of PBA fluxes for the entire validation year are shown in Fig. 9. These are high from Spring to early Autumn, when temperatures are favorable for microbial and plant growth and sharply decrease during Winter months in

response to a decrease in temperature and LAI, an increase in the mean wind speed accompanied by a decrease in the mean number of microorganisms populating the phyllosphere (Fig. 9). The model also predicts episodes in which the daily fluxes of microorganisms into the atmosphere are above and up to roughly twice that of the seasonal average. These events are often associated with persistent conditions of high wind and turbulence (Fig. 10a, 10b) and clear skies (Fig. 10c, 10d) which are typical of the synoptic weather conditions in Southeastern France, when high pressure in the Bay of Biscay and a low around

the Gulf of Genoa generate the wind that prevails from the north (called the "Mistral"). Under these favorable conditions microbial growth in the phyllosphere balances the high removal rates caused by turbulence, so that the overall microbial population on leaves sustains high transport (Fig. 10c).

## 4. Discussion

The results and tools we present here offer a new approach for studying bioaerosols. Previous attempts to understand the

distribution of PBAs in the atmosphere tended to simplify the surface-atmosphere transport both by deriving emissions from airborne concentrations (Burrows et al., 2009a) and by making ecosystem-wide assumptions about emissions (Burrows et al., 2009a;Hoose et al., 2010;Sesartic et al., 2012). Airborne concentrations are, nevertheless, variable, being the combined results of both emissive and depositional processes as well as atmospheric transport (Wilkinson et al., 2012) . For this reason, we conceived the PLAnET model to estimate fluxes directly, while accounting for the underlying emission-deposition processes.

We sought to capture the dynamics underlying microbial emissions, thereby making the airborne concentrations a direct consequence without further assumptions. The model tries to generate fluxes from the interactions of the phyllosphere population dynamics and the local meteorological conditions instead of employing only a regression framework from measured

data (such as in a previous attempt to simulate microbial fluxes Lighthart and Kirilenko (1998)). While both gross upward and downward flux in the PLAnET model are resolved separately, this does not happen when employing a gradient method, and the presence of depositional effects could affect the observed gradient. Following (Gillette et al., 1974;Gillette et al., 1997), depositional effect for particles < 10 μm are significant only when the ratio between the deposition and friction velocity is greater than 0.1.The value of this ratio did not exceed the critical thresholds either in the 2008-2010 campaigns, nor in 2015. This guarantee the applicability of the gradient method for the observations at the Montfavet site made in sufficiently turbulent conditions. Depositional effects were not relevant also in Park et al. (2011) when applying the gradient method to $PM_{10}$ fluxes. It has to be taken in account, though, that deposition depends on the particle diameter, and the choice of a fixed diameter for bioaerosols that was made here is a necessary simplification due to the impossibility of knowing the full size spectrum and its temporal variation. Seasonal variations in the size fraction containing most bioaerosols were in fact detected by Raisi et al. (2013).

The PLAnET deterministic framework follows the approach of Fall et al. (2016), which employed data from the literature on a specific pathogen, *Bremia lactucae,* to estimate its airborne concentrations. From Table 2 it is clear that the optimization procedure made a clear use of the imposed bounds in order to obtain feasible parameters. Not imposing feasible bounds would have posed a risk for the minimization to wander into physically unrealistic, but mathematically sound parameter space (i.e.: a set of parameters achieving a very small $\varepsilon$ by combining, for example, non-realistic growth temperatures). The optimal temperature chosen by the optimization algorithm for microbial growth, for example, is 21.6 °C. Considering that during summer days with higher vapor pressure deficit the leaf surface temperature can reach even a 5 °C difference from air temperature (Jackson et al., 1981;Wiegand and Namken, 1966) this would mean that the modeled optimal temperature is quite close to the incubation temperature used in the laboratory (25 °C). It is worth noting that, while a reasonable choice of growth temperature range was made for the overall microbial population, specific microorganisms may have different temperature optima. Future work can be done to fine tune such range on the species composition of the microbial source. The reliability of the optimization is backed up by the sensitivity analysis: any variation in the chosen parameters results in a worsening of the error statistic, as is clearly visible from Table 2

Compared to the model by Fall et al. (2016), PLAnET  falls short in terms of validation statistics. These differences can be explained by the different endpoints and scopes between the models. Since the PLAnET model aims to simulate an overall bioaerosol flux, instead of airborne concentrations of a single species as does the model by Fall et al. (2016), there are significant higher uncertainties involved in the process. Nevertheless, if the confidence intervals (CIs) for the slopes of Fig. 7 are taken into account, it can be seen that in 2008-2010 and in 2015, the 95 % CIs include 1 and exclude 0 (the 95% CIs are 0.36-1.05 and 0.41-2.21, respectively). This suggests that the main weakness of the model would only be the number of observations. Longer campaigns conducted on different ecosystems would help in better assessing the relationship between modeled and measured data as well as the "portability" of the PLAnET model to different ecosystems.

While the obtained results are quite promising, there are still some caveats to consider. One of the first improvements that would benefit the PLAnET model would be validation on microbial fluxes that are not based solely on cultivated

microorganisms. The culturable to total ratio may range from 0.01 to 75 % and is generally below 10 % (see Burrows et al. (2009b) and references therein), meaning that PLAnET output needs scaling to be compared with the work, for example, of Burrows et al. (2009a), Sesartic et al. (2012) and Sesartic et al. (2013). A simple comparison can be made between PLAnET simulated fluxes and fluxes reported in Burrows et al. (2009a) using the scaling factor for culturable-to-total bacteria for

grasslands (302, Burrows et al. (2009b)). An average total microorganisms flux of $750.5 \pm 1976$ cells $m^{-2}$ $s^{-1}$ was obtained for PLAnET that, although associated with a high variability, is similar to the median value for grassland (roughly 1000 cells $m^{-2}$ $s^{-1}$) reported in Burrows et al. (2009a). PLAnET can be used to predict microbial emissions and, when proper scaling factors are selected, it can be potentially used as a tool to link surface processes to the spatial and temporal dynamics of atmospheric processes since PBAs could represent an important component of the atmospheric aerosol load (at least at regional scales

Hummel et al. (2015)). There are only few quantitative, field-deployable sampling methods that target all microbial cells including non culturable ones. One of the most reliable that can be adopted for analyzing samples coming from different kinds of collectors is epifluorescence staining, which is able to discriminate biological versus non-biological particles in a sample independently from their culturability. Contrary to plate incubation, epifluorescence is able also to detect viable-but-nonculturable microbial cells, that is, organisms that are not dead, but are not in the condition of growing (see Oliver (1993)

and Burrows et al. (2009b) and references therein). The key issue with the method is the need for a minimum amount of particles per sample ($10^4$ per liter, Gandolfi et al. (2013)) and, therefore, an appropriate amount of sampled air. On rural sites Bowers et al. (2011) were able to employ epifluorescence sampling at $30\,l\,min^{-1}$ for 1.5 h, while Harrison et al. (2005) worked with high-volumetric sampling at $1000\,l\,min^{-1}$ for 6 hours at a time. Such timescales are not suitable for flux-gradient applications, where fluctuations in concentrations must be resolvable on a timescale appropriate for the PBL response time

($\leq$1 h). This is why in the present work Burkard samplers where chosen: both for their high volumetric flowrate ($500\,l\,min^{-1}$) and for their virtual impactor nature that is favorable for preserving particle viability. Still, in future studies, epifluorescence sampling performed on "relatively long" time intervals (e.g.: 1.5 h or more) could be used alongside more frequent (e.g.: 15 min) cultivable samplings to scale cultivable to total microorganisms, assuming that the culturability does not change in the longer time span. Ultraviolet induced laser fluorescence (UV-LIF) is a very recent methodology that measures PBAs

concentration from fluorescence emission and particles' characteristics, using statistical methods such as hierarchical agglomerative cluster analysis to distinguish between different types of PBAs (Crawford et al., 2015). UV-LIF has already been used to measure atmospheric PBAs (Huffman et al., 2010;Gabey et al., 2010) and, given its relatively fast response time, it has the potential to be used in combination with micrometeorological methods to estimate microbial fluxes. A first attempt in this sense has been made in a pine forest by Crawford et al. (2014). This method is very promising since it works

independently from microbial culturablity, even if research is still ongoing on discriminating between different PBAs classes and between PBAs and non-biological fluorescent compounds contaminating the signal (Gabey et al., 2013;Pöhlker et al., 2012;Toprak and Schnaiter, 2013). Single mass particle spectrometry (SPMS) is also a technique that can be used to detect PBAs by relying on the spectroscopic detection of specific compounds that are assumed as proxy of bioaerosols (Zawadowicz et al., 2017). Similarly to UV-LIF, this method does not rely on PBAs culturability  while suffering from interference of non-

biological particles having coincident spectral peaks (Zawadowicz et al., 2017). It is important to consider, though, that even if live and dead microorganisms would contribute to cloud-related processes due to their chemical and physical composition, the latter would not matter from an evolutionary perspective. Live cells have a chance of further transmitting their biological and chemical footprints to a wider microbial population. The latter would represent an ecological feedback increasing the

population of live particles with characteristics that could favor survival and, eventually, physical interaction with atmospheric processes (i.e.: increased expression of given proteins). While it is true that estimating fluxes of total biological particles is important from a biogeochemical point of view Burrows et al. (2009a), measuring the viable fraction of fluxes would give information about the amount of microorganisms that can potentially survive transport. The second critical improvement would be to validate and test the model on data from a larger number of different ecosystems, such as forests and agricultural crops.

Another caveat regards the parametrization of deposition. The simplified version of the model of Slinn (1982) implemented in PLAnET does not take into account the presence of potential negative gradients between atmosphere and canopy, which was not possible to investigate during the present sampling campaigns. This aspect needs further investigation for a better representation of particle deposition in such conditions.

The prognostic capability of the model has been investigated by running the model between 2001 and 2015. In the years where

meteorological data from Montfavet were not available (2001-2006), we used the average of the closest four points from the Climate Forecast System Reanalysis hourly time series (Saha et al., 2010). Season-averaged net fluxes showed small inter-annual variation in winter when fluxes fluctuated around zero (-0.01 to 0.25 CFU $m^{-2}$ $s^{-1}$) versus higher and more variable average fluxes in summer ranging from 3 to 8.3 CFU $m^{-2}$ $s^{-1}$. The model was able to represent the interplay of the different meteorological variables: summer 2003 was characterized by one of the lowest average phyllosphere population sizes due to

the exceptionally high temperatures registered during that year hindering microbial growth. In fact, in summer 2003 the average population size was 2.85 x$10^5$ CFU $m^{-2}$ versus a seasonal average of 3.86 x$10^5$ CFU $m^{-2}$ and an average temperature of 26°C versus 23°C. The lowest average summer population size was simulated, instead, in 2006 (2.66 x$10^5$ CFU $m^{-2}$), when an unusually low LAI (0.8 versus 0.91 inter-annual seasonal average) influenced the maximum amount of microorganisms that were able to grow along with a high average friction velocity increasing microbial removal (0.33 versus 0.25 m $s^{-1}$ inter-annual

seasonal average). Still, there are some potentially unresolved meteorological forcings that the model does not take into account, such as rainfall. Huffman et al. (2013) and Prenni et al. (2013) provided convincing experimental evidence that bioaerosol and ice nucleator concentrations increased during and shortly after rainfall events. Rainfall, in fact, may boost PBA emissions. On the one hand this may be due to the impact of raindrops that shake plants generating a detaching force (McCartney, 1991;Robertson and Alexander, 1994) and on the other hand it may be due to a boost in growth and a subsequent

increase in the concentration of PBAs on the plant. For example, for *P. syringae* a 35-fold increase was seen after 48 hours from a precipitation event (Hirano et al., 1996). This would increase the population of particles available for transport. This seems to contrast classical wet scavenging theory, where falling hydrometeors deplete the atmosphere of suspended particles (Seinfeld and Pandis, 2012), and, therefore, act as a sink for particulate matter (see for example Tai et al. (2010) or Ouyang et al. (2015)). Due to this complex interaction between PBAs and rainfall, precipitation and humidity were not taken into account

in this first version of the PLAnET model. In fact, if a very rainy month (September 2010 with an average of 0.13 mm of rain per hour) is compared against a non-rainy month (September 2007 with an average rainfall of 0.027 mm $hr^{-1}$), average net fluxes are very similar and actually greater in the less rainy month (4.87 CFU $m^{-2}$ $s^{-1}$ in September 2007 versus 4.35 CFU $m^{-2}$ $s^{-1}$ in September 2010).

But rainfall is not the only process that can influence net fluxes: intensive grazing, mowing and harvesting are also activities that can impact bioaerosol emission from a grassland. Such effects, however, are not straightforward and not completely known. Intensive grazing, for example, would damage the canopy, affecting LAI and the available population. On the other hand, by damaging plants and releasing nutrients from plant tissue it could enhance microbial growth on leaves. Furthermore animals themselves are potential bioaerosols sources (animal manure contains a large variety of microorganisms Cotta et al.

(2003)). Harvesting would also contribute to the reduction of the source term (reducing LAI), but, along with many agricultural operations, it can generate a higher amount of suspended particles (see for example Hiscox et al. (2008)), potentially containing bioaerosols. All of these complex interactions can therefore generate both transient and lagged effects, which are still not taken into account by any model and should be investigated in the future.

## 5. Conclusions

This study investigated with multiple campaigns the behavior of a Mediterranean grassland from the point of view of microbial emissions. Across the campaigns fluxes of microorganism have been estimated through a sound micrometeorological method (flux-gradient methodology). The applicability of the method was assessed by comparing water vapor gradient fluxes with those measured directly by eddy covariance: the lack of significant divergence between the two, suggests that the gradient methodology was applicable in the experimental conditions, even if it needs to be acknowledged that the good correspondence

in terms of water vapor fluxes does not necessary apply to bioareosol flux measurements. Bacterial and fungal colonies, in fact, behave as a passive tracer in the absence of significant aerodynamic effects: such condition was therefore tested during field campaigns using literature relationships between friction velocity and size of transported particles. The grassland showed a majority of emission fluxes , with a magnitude comparable to what was previously seen on a desert scrubland (Lighthart and Shaffer, 1994). The collected data were used to calibrate and validate a deterministic model (PLAnET) for estimating emission

of microorganisms from the surface. Even if there are still some open issues in the model (namely the relationship between flux, precipitation and the culturability of microorganisms), PLAnET provides previously unavailable insights into the dynamics of microbial fluxes and the underlying driving forces. Hopefully its evolution within the scientific community will be fostered not only by its ease of use (few easily accessible meteorological parameters are needed for its operation), but also by the robust framework for estimation of microbial fluxes (the model code can be freely downloaded at

https://it.mathworks.com/matlabcentral/fileexchange/63257-planet-microbial-model). Its computational simplicity makes it also an attractive addition to larger scale models aiming at simulating the dispersion of PBAs at regional or global scales (such as CALPUFF, Scire et al. (2000); WRF-CHEM, Grell et al. (2005); CHIMERE, Menut et al. (2013); ECHAM-HAM, Stier et

al. (2005) or CAMx). The PLAnET model can in fact be an emission module receiving the same meteorological forcing of the dispersion model within which it is nested and generate time-varying emission rates at surface level. PBAs could be added as a chemical species with given characteristics, in addition to all the aerosols coming from biogenic and anthropogenic inventories. The larger-scale model would be then able to simulate their dispersion across the simulation domain giving new insight in the potential of PBAs to impact precipitation, as well as exploring different scenarios about the potential pathways of transport of plant pathogens from the phyllosphere. In fact, given that some of the aforementioned models are able to simulate both gas-phase and aerosol chemistry, it would be possible to follow up the pioneering work of Burrows et al. (2009a) and Sesartic et al. (2013) at different spatiotemporal scales and investigate the changes in the outputs due to the insertion of a realistic PBAs emission module.

## 6. Acknowledgements

Part of this work was funded thanks to the AirFors project (PIAPP-GA-2011-286079) part of the Seventh Framework European Programme. The authors wish to thank Alessandro Zaldei for his contribution to the preparation of the instrumentation for the 2015 campaign, as well as the technical and engineering staff from the INRA Unit of Plant Pathology (Jean-Marc Bastien, Joël Beraud, Jean-Francois Bourgeay, Frédéric Pascal and Michel Pascal) for their support during the field campaigns.

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

| Parameter | Range Boundaries | References for Ranges | Units |
|---|---|---|---|
| $T_{MIN}$* | 5-15 | Standard Mesophilic Range | °C |
| $T_{MAX}$* | 30-45 | Standard Mesophilic Range | °C |
| $T_{OPT}$ | $T_{MIN}$-$T_{MAX}$ | Standard Mesophilic Range | °C |
| c* | 0.1-2 | Inserted by the authors | None |
| $k_{min}$* | $4.7 \times 10^4$ - $4.7e \times 10^5$ | Hirano and Upper (1986); Wilson and Lindow (1994) | CFU m$^{-2}$ |
| $k_{max}$* | $4.7 \times 10^5$-$4.7 \times 10^8$ | $k_{min}$ and Hirano and Upper (1986) | CFU m$^{-2}$ |
| $m_1$* | 22.3-35 | Derived by the authors from Lighthart and Shaffer (1994) | CFU m$^{-2}$ s$^{-1}$ |
| $m_2$* | 250-260 | Derived by the authors from Lighthart and Shaffer (1994) | CFU m$^{-2}$ s$^{-1}$ |
| $m_3$* | 17-23.3 | Derived by the authors from Lighthart and Shaffer (1994) | None |
| $\xi$ | 1800 | Equation (10) | Seconds |
| $p_1$ | 26.99 | Derived by the authors | CFU m$^{-3}$ |
| $p_2$ | 115.9 | Derived by the authors | CFU m$^{-3}$ |

**Table 1:** Estimates for model parameters. The parameters tagged with an asterisk (*) are those that entered the calibration as unknowns from the initial guess, while the other parameters were fixed.

| | $T_{MIN}$ (°C) | $T_{MAX}$ (°C) | c | $k_{min}$ (CFU m$^{-2}$) | $k_{max}$ (CFU m$^{-2}$) | $m_1$ | $m_2$ | $m_3$ |
|---|---|---|---|---|---|---|---|---|
| Value | 12.96 | 30.16 | 0.13 | $5 \times 10^4$ | $4.82 \times 10^6$ | 30 | 256.26 | 19 |
| Sensitivity | 0.65 | 0.29 | 0.23 | 0.03 | 0.23 | 0.35 | 0.08 | 0.47 |

**Table 2**: Results of the optimization procedure and the sensitivity analysis. The first row reports the value chosen by the optimization for the parameters in the column headers, while the second row reports the sensitivity value for each parameter.

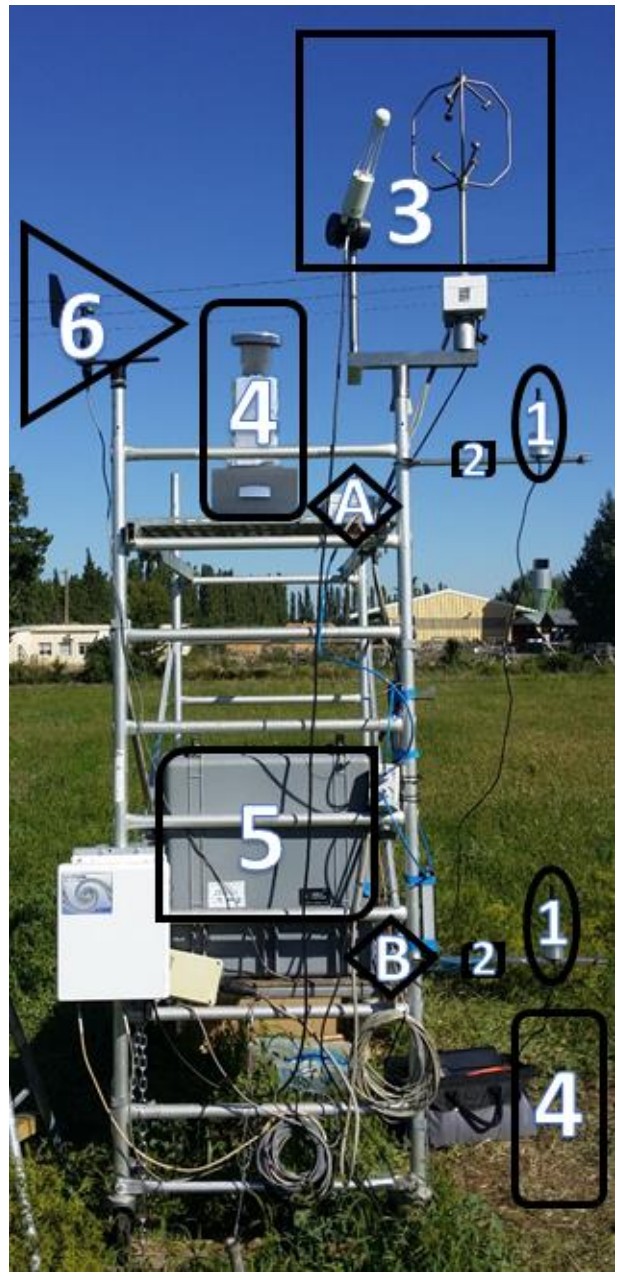

**Figure 1:** Schematic representation of the sampling station: each piece of equipment is represented by a number (in bold face) and the position of the equipment, expressed as cm a.g.l is indicated in parenthesis. Cup anemometers (**1**; 80 and 210), thermocouples (**2**; 80 and 210), sonic anemometer and Li-7500 open-path gas analyzer (3; 300), Burkard air samplers (**4**; 75 and 255, the bottom one is not in place in the figure, but the rectangle indicates its approximate position), Li-7000 differential gas analyzer (**5**; with inlets at 55 A and 200 B) and wind vane (6; 250).

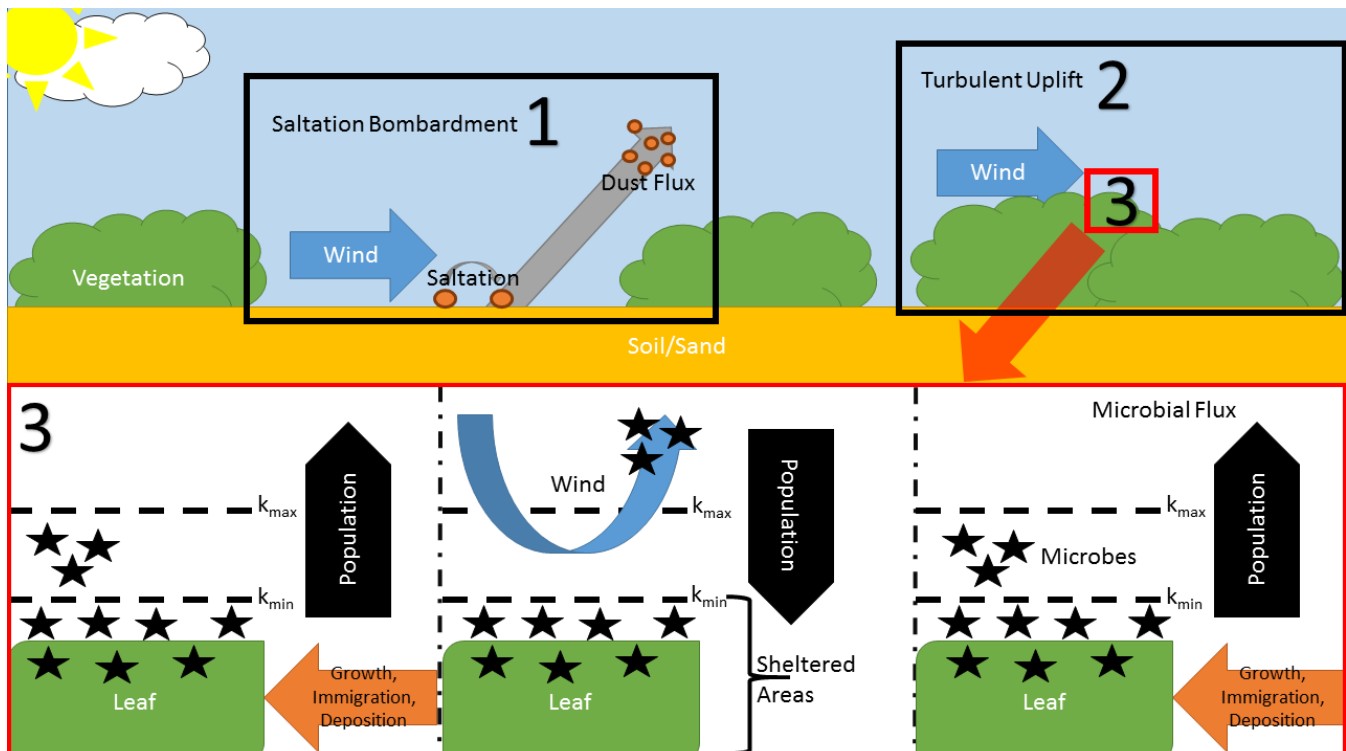

**Figure 2:** Schematics of the differences between Aeolian dust flux and microbial flux on which the PLAnET model is founded. Box 1(top left) shows the typical dust saltation mechanism. The action of wind (blue arrow) on soil makes dust particles (orange dots) "jump" for short distances, ejecting smaller dust particles in the atmosphere. Turbulent uplift is shown in Box 2 (upper right) where wind acts on the phyllosphere. What happens in the small red box 3 is zoomed in into the lower part of the figure (as indicated by the red arrow). Phyllosphere harbors a given amount of microbial particles (black stars) up to a maximum (carrying capacity, $k_{max}$ in the figure). Wind can remove a certain fraction of "available" microorganisms up to the limit of a sheltered fraction of the population ($k_{min}$ in the figure). While the action of wind decreases the amount of particles on the leaf, the population keeps experiencing phenomena such as growth, immigration from other leaves and deposition of airborne particles, all contributing to an increase in population. The balance between population dynamics and uplift is what contribute to the net flux simulated by the PLAnET model.

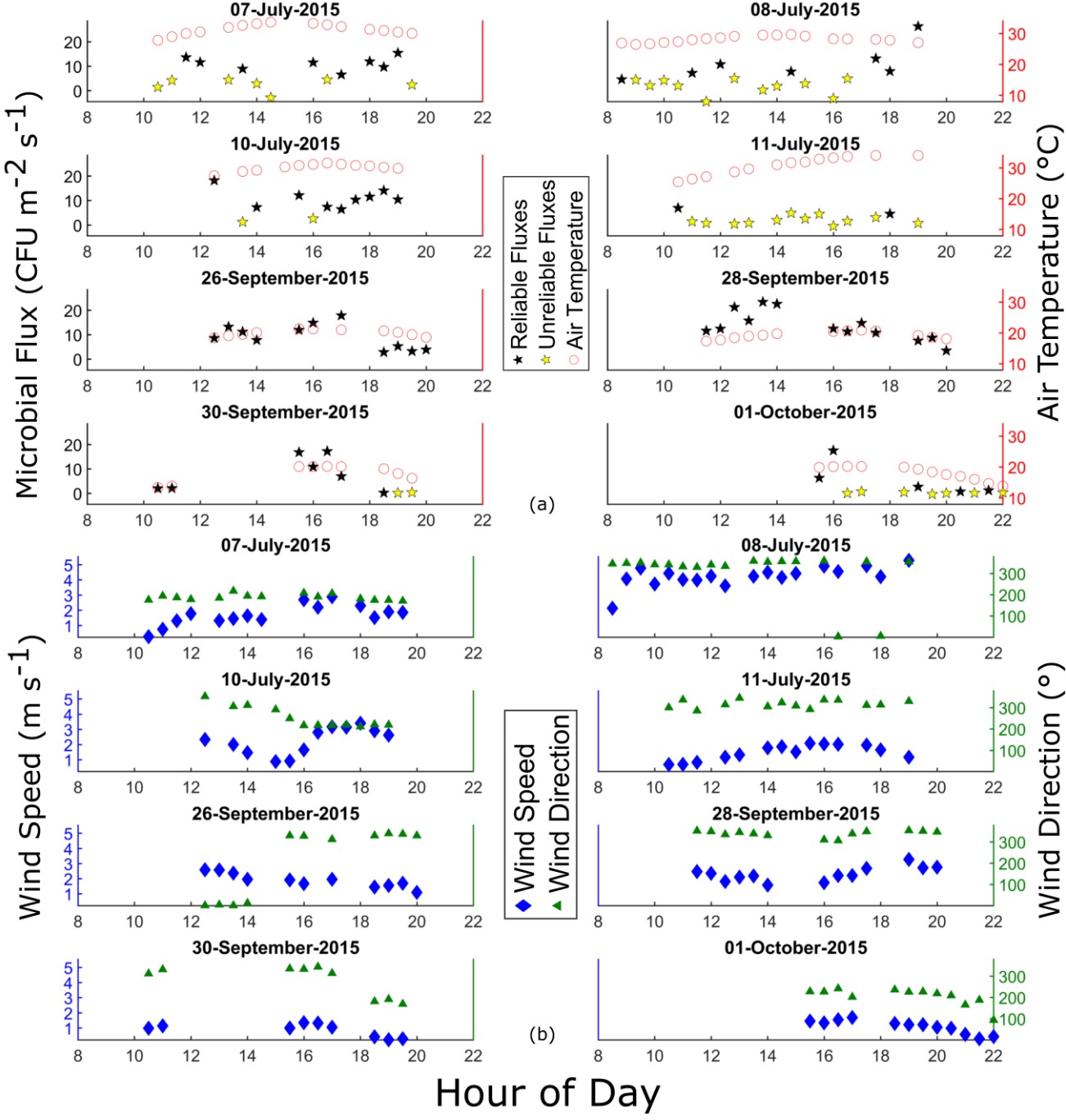

**Figure 3**: Dynamics of observed microbial fluxes, air temperature, wind speed and wind direction in the 2015 field campaigns. Fig. 3a shows the time series of air temperature and microbial fluxes for July and September-October 2015. Unreliable fluxes are those below the MRG. Fig. 3b shows the time series of wind speed and wind direction for the same periods.

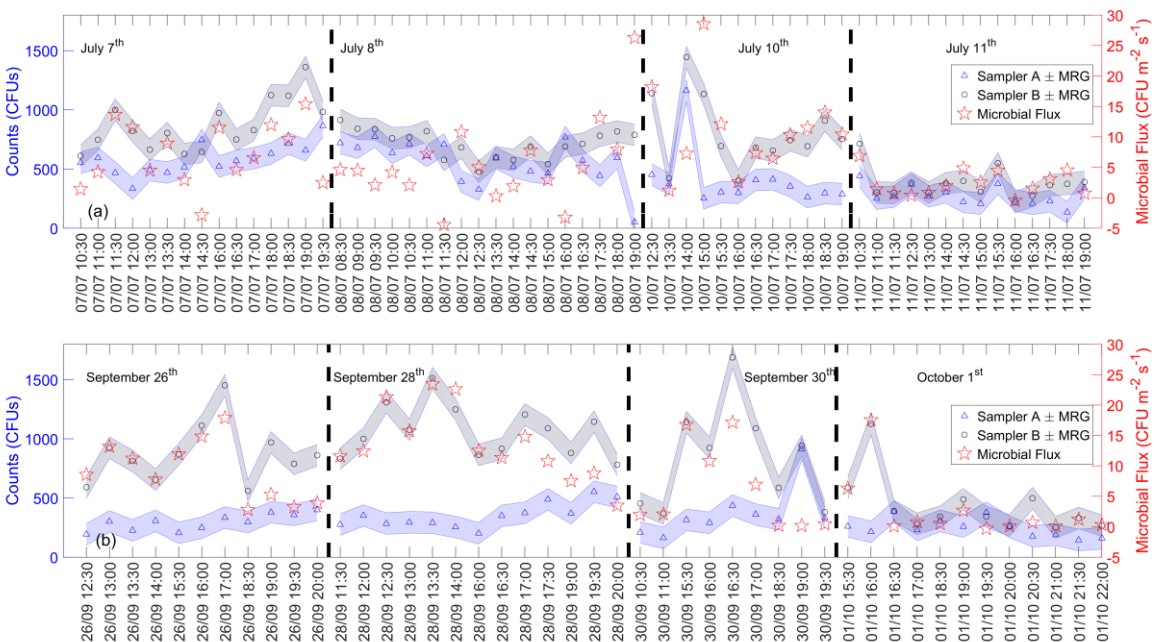

**Figure 4**: Details on the relationship between counted CFUs by the Burkard samplers and the estimated fluxes for the campaigns of July (a) and September (b) 2015

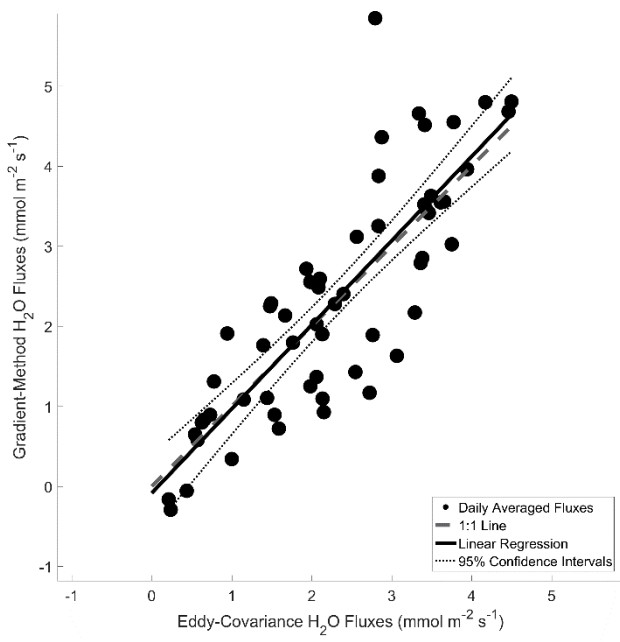

**Figure 5**: Water vapor fluxes measured via the eddy covariance method (x-axis) *versus* those derived from the flux-gradient method (y-axis). The plotted linear regression has a slope of 1.05, an offset of -0.08 and explains 70% of the data variability.

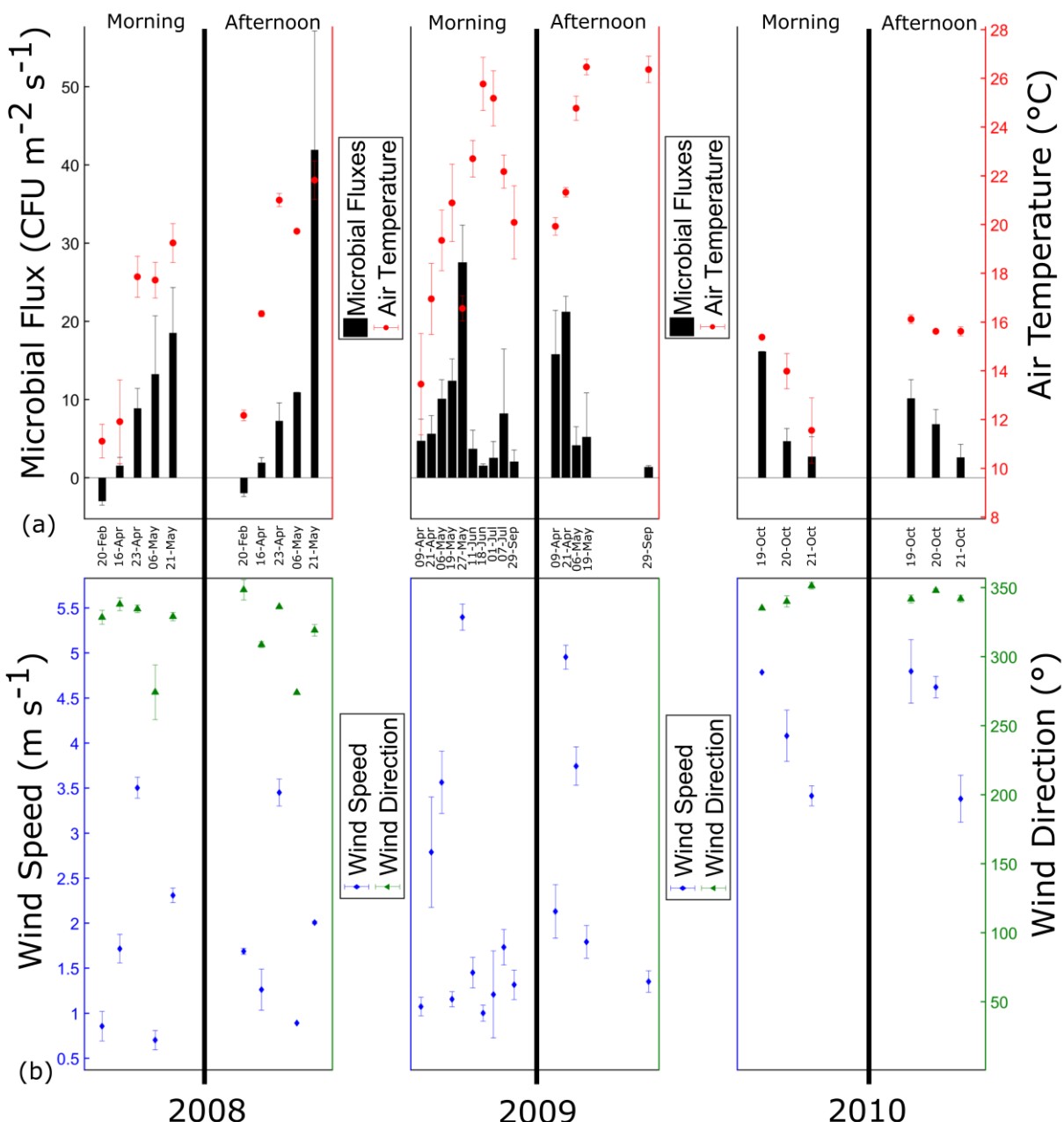

**Figure 6:** Temporal dynamics of observed microbial fluxes, air temperature, wind speed and wind direction in the 2008, 2009 and 2010 field campaigns. Fig. 6a shows time series of air temperature and microbial fluxes for the campaigns held between 2008 and 2010. Fig. 6b shows the time series of wind speed and wind direction for the same period. No wind direction data were available for the year 2009. For both figures 6a and 6b morning and afternoon averages are reported with the relative standard error in the error-bars.

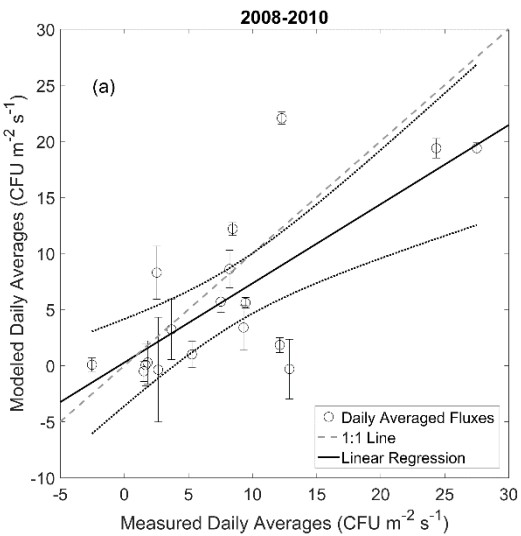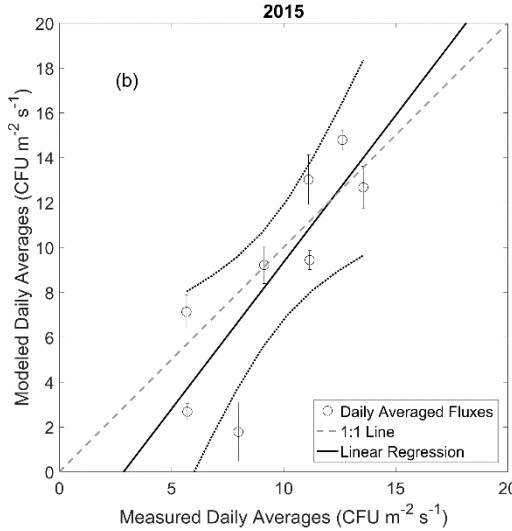

**Figure 7:** Relationship between measured and modeled daily averages of microbial flux. Fig. 6a shows the regression between the daily averages from the 2008-2010 campaigns and the optimized model (y=0.70x+0.28; $r^2$=0.54) and Fig. 6b shows the one from the 2015 campaigns and the model (y=1.31x-3.75; $r^2$= 0.68). The error bars are derived from the following equation $\pm\sigma\left(\frac{m_i-o_i}{\overline{m_i,o_i}}\right)$. Where σ is the standard deviation of the ratio between the difference between modeled and observed points $(m_i - o_i)$ and the average of the same points $(\overline{m_i, o_i})$.

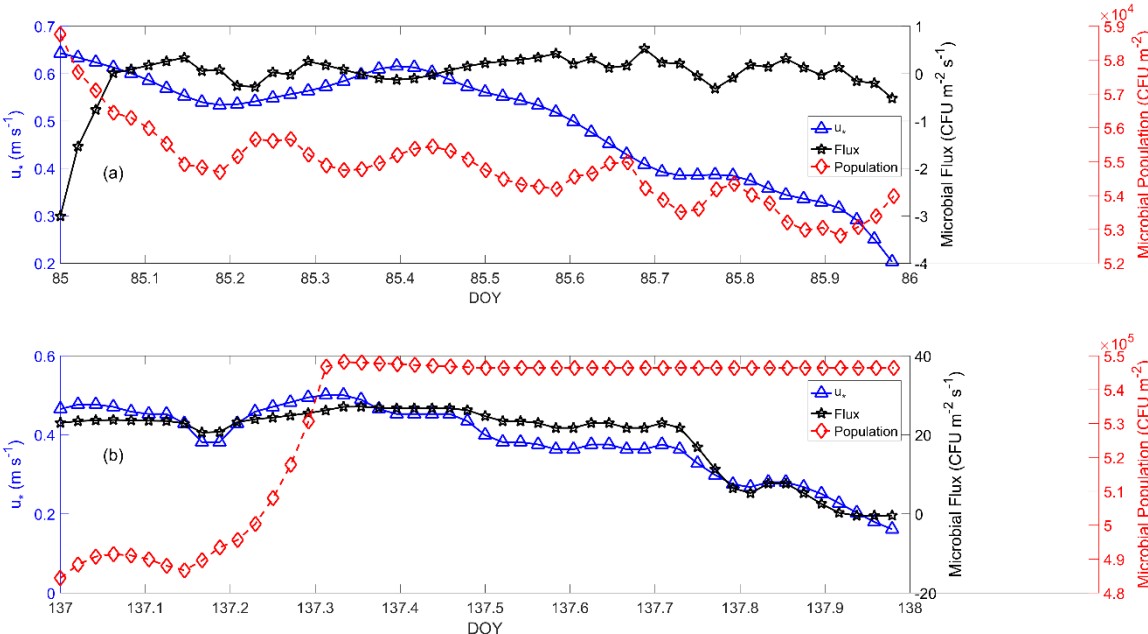

**Figure 8:** High u∗/low flux event (a) and low u∗/high flux event (b) as simulated for the year 2015. The x-axis indicates the fractional day of the year (DOY, northern hemisphere) for these two events, the left y-axis shows the u∗ values, the first right y-axis shows the microbial net flux, and the second right y-axis shows the phyllospheric population size.

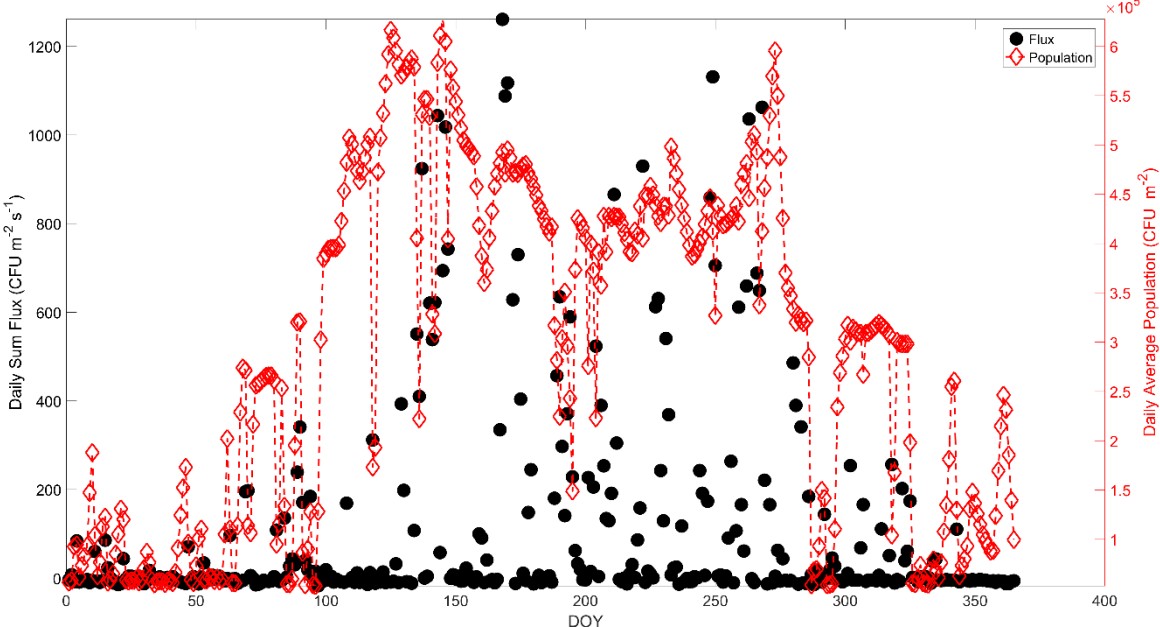

**Figure 9**: Time series of the daily sum of modeled microbial fluxes and daily averages of the surface microbial population for 2015. The labels on the x-axis define the DOY starting from 1 January 2015 (DOY 1, northern hemisphere winter) to 31 December 2015 (DOY 365, northern hemisphere winter).

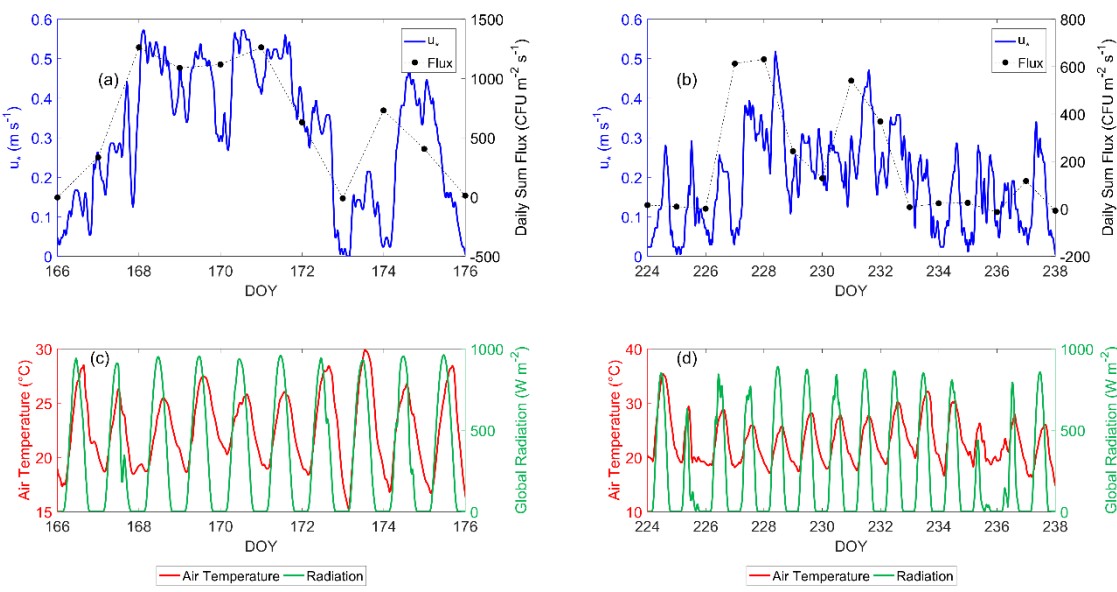

**Figure 10**: High-wind events in 2015 in Montfavet, France. Plots a) and b) show friction velocity (left y-axis) and the daily sum of flux (right y-axis) for two high-wind events (DOY 166-176 and 224-238 of 2015). Plots c) and d) show solar radiation (right y-axis) and air temperature (left y-axis) for the same two events.

