# Peer review of "Measurements and modelling of surface-atmosphere exchange of microorganisms in Mediterranean grassland"

_Atmospheric Chemistry and Physics, 2017_

## Referee Comment (RC1) · Anonymous Referee #1 · 13 Jul 2017

1) General Comments The submitted manuscript presents both a micro meteorological measurement system as well as a modeling framework for time resolved quantification of biological aerosol and microorganism ecosystem-atmosphere exchange fluxes. Observational data was obtained over a Mediterranean grassland site in France over 3 measurement campaigns. A deterministic modeling framework was calibrated based on the first 2 measurement campaigns and model performance as well as sensitivities to input variables were evaluated on a more recent independent measurement campaign. In both domains the authors succeeded in presenting a conclusive and novel

combination of methods, producing a new (and to my knowledge the most directly measured) data set of high relevance. The proposed PLaNET model combines physically sound expressions of phyllosphere population dynamics and turbulent removal processes and was optimized through heuristic/trainable algorithms for solving non linear functions. Especially due the open source nature of their model and a detailed error and sensitivity analysis, both model and data set can be used and updated in the future. The manuscript accomplishes to reveal important insight (e.g. non linear responses) into interactions between bacteria/fungi population dynamics, grass land micro meteorology and the resulting microbial exchange fluxes. Therefore I recommend publishing this work in ACP following minor revisions.

2) Specific Comments

P2L22: "...some periods...some land uses." Rather vague description doesn't help the reader in getting an overview of previous work, rephrase or delete. P2L32ff: "But while Burrows ..." The whole sentence should be rephrased to improve readability. The word "species" is used but it's not entirely clear if the authors mean microbial species. Also it is unclear why "in reality" emissions are different from the results from Burrows et al.. If this is a conclusion based on the submitted work, it should rather be made in the conclusions chapter. P3L13ff: The site description is too general (e.g. no species resolved vegetation or bare soil coverage). This is surprising given that the authors describe the existence of a large variability in emission fluxes "especially because of variation in vegetation across space (P3L3)". Was the grassland actively managed, e.g. grazed or mown during or in between sampling? Was the management comparable between different campaigns and study sites? P3L24: How does the flow rate translate to Reynolds Numbers. What are the resulting losses in the sampler's intake? Are losses biased towards larger or smaller aerosols? P3-4 Chapter 2.1: General questions regarding the employed micro-meteorological measurement technique: a) Besides precise instrumentation for the concentration gradient, steady state conditions are the key restriction for applying k-theory/gradient measurements. Was the sonic

Interactive
comment

data used to investigate steady state conditions, e.g. employ standard quality checks: stationarity etc. (Foken and Wichura, 1996)? b) Results from detectors precision study (MRG) are visible in figure 3 only. It would be helpful to also present actual values and compare them to measured fluxes. c) was scaling of measurement height by the zero-mean displacement height discarded in estimation of K due to the comparably small canopy? d) Since the authors employed a fast 3d sonic anemometer, I would welcome the addition of a more data driven flux footprint and possible flow distortion evaluation besides the cited literature relationships the authors followed. Did you discard measurements from specific wind directions, e.g. situations where sensor inlets are located downwind from the tower structure? P4L23: "similar herbaceous species, such as..." these species were not listed for study site 1. P5L7: You assume that deposition is purely driven by gravitational settling? Are other means (e.g. negative gradient between vegetation canopy and atmosphere, interception, impaction) insignificant? P5L18: Why did you use the Lighthart and Shaffer data to express observed fluxes as a logistic function of ustar? What is the goodness of fit for m1, m2, m3 and eq.2 in general? How does the fitting errors propagate in overall model uncertainty. P6L5: The settling velocity is highly dependent on particle diameter and shape. Obviously for the applicability of a model, simplifications have to be made here, since time resolved aerosol size and density spectra are hard to measure or predict. However it would be worth exploring if deposition (i.e. Vg) has a larger impact on overall predicted net fluxes by varying aerosol size modes and densities (in reality these will have temporal patterns for instance due to phenology of different sources). It should at least be specified if the used literature values for particle density and particle diameter are representative only for grasslands or a specific season. P6L10: eq. 6: please report goodness of fit for linear regression between avg C and LAI. How does the uncertainty in Ca propagate into prediction of Fd? P8L5: The comparison between flux gradient and eddy covariance flux measurements provide confidence in the observations. However, eddy fluxes are not necessarily the truth, flux errors in LE are often in the magnitude of tens of percent. EC measurements were made at different height than

the gradient measurements, meaning that the EC instrumentation sees different parts of the grassland. Along these lines, the open path EC sensor was not cross-calibrated with the closed path gradient sensors? In other words, It will be hard to conclude if the gradient or the EC results are off. A more detailed error/uncertainty discussion of the net fluxes obtained from the gradient method would be appreciated here. It should at least be acknowledged that the conclusions made from the H2O flux comparisons do not necessarily apply to aerosol flux measurements. What are the expected uncertainties introduced through assuming scalar similarity? The MRG or precision of the detectors employed at same height could be used to propagate a flux uncertainty. Since the model is calibrated on the measured fluxes (plus minus uncertainty) also the model will have this uncertainty. P8L15ff: the 2008/2009 measurements have a wider spread, partly due to the fact that no detection limit was applied (e.g. in 2015 all negative fluxes were removed due to the detection limit). What is the reason for that? P11L33ff: besides rainfall other events could have an effect on PBA production. You mentioned the heat wave in 2003. What about water stress or cutting/mowing/ grazing. Some of these stress factors might have lagged interactions with LAI and microorganism population growth. It would be great to introduce 1 or 2 sentences about these effects and how they would change the annual emission from a grassland, if feasible.

Fig(3): Why is the detection limit (MRG) half in Sept-Oct as compared to July?

3) Technical Corrections: P1L19: ". . .than that. . .". Numerus P3L12: "similar terrain". Phrasing: Do you mean flat? P8L7: delete ",instead," P8L7: rephrase, e.g.: spanned over multiple seasons P9L32: ". . . all the emission-deposition processes", please rephrase P12L14: ". . .has been". Numerus P12L21: ". . .outward fluxes". Rephrase, e.g. emission fluxes

Foken, T. and B. Wichura. 1996. Tools for quality assessment of surface-based flux measurements. Agricultural and Forest Meteorology, 78: 83-105.

---

## Referee Comment (RC2) · Anonymous Referee #2 · 28 Jul 2017

The manuscript presents analysis of flux measurements and development of a model to simulate the emissions of culturable microbes from Mediterranean pasture. High uncertainties exist in the knowledge about the emissions and abundance of primary biological particles in the atmosphere and the development of an emission model for microbes is a step forward in closing this gap. The strength of the model is its ability to simulate the dynamics of the microbial population in the vegetation, and use that information together with meteorological factors to simulate the emissions to the atmosphere. However, the model currently has a few shortcomings, also admitted by the

authors, that make it not immediately useful for large scale applications. Firstly, the model is calibrated using the observations of culturable microorganisms. However, this is not what is needed for the most probable uses of the model mentioned by the authors in the abstract. Studying the spread of pathogenic species needs emissions of those pathogenic single species, while most applications in biogeochemistry would require the emissions of total biogenic aerosol, not only the culturable species. Secondly, the model has only been calibrated for Mediterranean grassland. However, this is still a big step forward in the field of modelling microbial emissions, so although further development is needed before the model becomes usable as an emission module for large scale atmospheric dispersion models, I fully support this work to be published in ACP.

I have a few comments and questions.

Discussion: Please include comparison between the PLAnET estimates and the previous studies, e.g. the Burrows et al, (2009b) and microbial flux observations from other locations, for instance using the same scaling factor to total microbes as Burrows et al. (2009a) used for grasslands (302).

Page 11, first paragraph suggests that obtaining a scaling factor to total microorganisms from the culturable fraction requires flux measurements of the total microorganisms. Why cannot the scaling factor be estimated from lower temporal resolution concentration measurements of both total microbes and the culturable fraction in the same site the flux measurements are made?

Page 11, lines13-18: Culturable and viable are not necessarily the same thing (see e.g Burrows et al, 2009a)

Minor remarks: Page 2, line 20-23: "In the past, only few attempts have been made to quantify the flux of microorganisms from plant canopies (Lindemann et al., 1982; Lindemann and Upper, 1985; Lighthart and Shaffer, 1994) covering only some periods and some land uses." This is confusing, as later the authors themselves reference several other studies that have tried to quantify the fluxes using various methodology.

[Figure]

Please specify that this sentence refers to direct measurements of bacterial fluxes.

Page 4, line 23-24 "The experimental field for these previous campaigns was covered with similar herbaceous species such as cocksfoot (Dactylis glomerata), ryegrass, tall fescue (Festuca arundinacea) and alfalfa (Medicago sativa)." The list of species for the other field on the previous page only included clover and ryegrass, so how similar was the vegetation on these fields?

---

## Referee Comment (RC3) · Anonymous Referee #3 · 28 Jul 2017

**General comments**

Carotenuto et al. investigated the ecosystem-atmosphere exchange of microorganisms in a Mediterranean grassland using flux-gradient measurements and a model that simulates biological production of microorganisms and the meteorological drivers of their emissions into the atmosphere. The presented research is of great interest because while concentration measurements of bioaerosols are getting relatively common, flux measurements are much rarer. Further, they designed an emission model to estimate the net flux of microorganisms from the phyllopshere to the atmosphere, which simulates the biological production of microorganisms and the meteorological divers of their emission. However, I have a number of serious concerns with the model that should be addressed before the paper can be considered for publication.

**Specific comments**

Some equations contain errors and units are missing or incorrect for a number of parameters, see the specific comments below. There are also a number of inconsistencies between the equations in the MS and in the code. I am puzzled most by the formulation for microbial population growth: is it assumed to respond instantaneously to changes in driving variables (temperature)? If so, is that a valid assumption on the 30 min. time step that you applied here, and at which time step would this assumption brake down? Or are the dynamics of the microbial population calculated transiently? Moreover, the formulation and units of eq. 8 are inconsistent, which is where most of my confusion comes from.

Why is only gravitational settling considered as removal mechanism? Other dry deposition mechanisms can be relevant for particles of the assumed size (3.3 um). How sensitive are the calculated dry deposition fluxes to assumptions on the particle diameter?

The title promises new insights into microbial fluxes, but I do not see them in the abstract or conclusions. What are for instance the 'underlying driving forces (P12,L25)' of microbial emissions? What new insights has the combination of the flux measurements and the emission model yielded into these driving forces? Could you highlight these findings in the abstract and conclusion?

I would like to see some more discussion on which types of microorganisms are sampled. The MS mentions viable microorganisms. Does that include both bacteria and fungal spores? Besides, can you say something about the size range of the observed particles? This will be important to eventually evaluate the role of the emitted bioaerosols on climate.
P2,L21: in addition to these papers, (Crawford et al., 2014) measured PBA fluxes using the flux-gradient method, and (Ahlm et al., 2010; Whitehead et al., 2010) measured fluxes of coarse aerosol in tropical forests (presumably PBAs) using eddy-covariance

P5,L2: competition is mentioned here as a driver of the microbial dynamics, but I don't think it is actually included in the model. Please limit this description to processes that are included in the model.

P5,L30: why is only gravitational settling included? For supermicron particles, also inertial impaction is important.

P6, eq8: I have some serious concerns regarding this equation, both as presented in the MS as in the code; this equation has microbial population size in the same units as the microbial growth and emission flux, which cannot be true. Should it read dN/dt=rN-Fn? In that case, it would represent exponential growth of the microbial population and loss due to emission. In the code, it is implemented as N(t)=N(t-1) + N(t-1)\*r + Fn, in which units are also inconsistent. It could be solved by multiplication of the 2nd and 3rd term on the RHS by the timestep, which would yield a discretization of the equation for exponential growth.

P6,L24: it is unclear what is meant here: 'kmin, which is the point at which all process find an equilibrium'

P7, L14: can you discuss how this choice has affected your results? This number seems to be important in determining the upper and lower bounds of the modeled microbial population.

P9,L28-31: strictly spoken, the Burrows et al 2009a study does not discuss the effect of PBAP on precipitation, which is what this sentence seems to imply

P9,L32: I would add transport to 'emission-deposition process' (e.g. Wilkinson et al., 2012)

P10,L28: what does it mean if the 95% confidence intervals include 0 and 1 or not?
P11,L2-12: I miss a discussion here on the use of online detection of PBAs using fluorescence measurements (e.g. Gabey et al., 2010; Huffman et al., 2010) or single particle mass-spectrometry (Zawadowicz et al., 2017). These techniques measure concentrations, but could in principle be used in combination with micrometeorological techniques to measure fluxes (e.g. Crawford et al., 2014).

P11,L18: it is unclear what is meant here: 'it is not to underestimate the long-term importance of evaluating the viable fraction of said fluxes'. Please rephrase

P12,L9-10: is rain rate given in mm/hour here?

Fig. 6: with half-hourly observations and model data available, why are only daily average fluxes given? In addition, it would be interesting to see time series of observations and model data.

**Technical issues**

P4,L15: unit is missing for z0

P5, eq 2: in the code, Nk\_max is given as N/k\_max, which seems correct to me, as it would express the population scaled by the carrying capacity, and judging by the units. Besides, the values of m1-m3 differ slightly from those in L19. What are the units of m1-m3? They cannot all be unitless (as mentioned in Table 1 and 2) when Fe is in CFU m-2 s-1.

P6, eq 9: this equation seems to be missing an exponent ((Topt-Tmin)/(Tmax-Topt)), which is included in the code. What is the unit of r? Based on eq. 8 it should be s-1. Then also c should have this unit, and not none, as mentioned in Table 1 and 2. Please check these and other units throughout the MS.

P9,L9: won't -> will not

P10,L23: remove 'it'

P10,L24: the Planet -> Planet
P11,L1: a scaling -> scaling

P11,L14: transmit -> transmitting

P11,L15: represents -> represent

P11,L16: the atmospheric -> atmospheric

P12,L6: acting -> act

P12, L32: which is nested -> which it is nested

P12,L14: has-> have

P12,L24: suggest adding a comma between 'precipitation and'

P12,L32: which is -> which it is

Fig. 3 and 5: Data within years are plotted as if they represent time series (with continuous lines), but this is not always the case. This makes the plots hard to interpret. Besides, time labels are placed at irregular intervals. Pls update these figures to make them easier to understand.

In the code at L305: in the Cc calculation, a factor of 2 is missing in the exponent

References

Ahlm, L., Krejci, R., Nilsson, E. D., M\a artensson, E. M., Vogt, M. and Artaxo, P.: Emission and dry deposition of accumulation mode particles in the Amazon Basin, Atmospheric Chem. Phys., 10(21), 10237-10253, doi:10.5194/acp-10-10237-2010, 2010.

Crawford, I., Robinson, N. H., Flynn, M. J., Foot, V. E., Gallagher, M. W., Huffman, J. A., Stanley, W. R. and Kaye, P. H.: Characterisation of bioaerosol emissions from a Colorado pine forest: results from the BEACHON-RoMBAS experiment, Atmospheric Chem. Phys., 14(16), 8559–8578, doi:10.5194/acp-14-8559-2014, 2014.
Elbert, W., Taylor, P. E., Andreae, M. O. and Pöschl, U.: Contribution of fungi to primary biogenic aerosols in the atmosphere: wet and dry discharged spores, carbohydrates, and inorganic ions, Atmospheric Chem. Phys., 7(17), 4569–4588, doi:10.5194/acp-7-4569-2007, 2007.

Gabey, A. M., Gallagher, M. W., Whitehead, J., Dorsey, J. R., Kaye, P. H. and Stanley, W. R.: Measurements and comparison of primary biological aerosol above and below a tropical forest canopy using a dual channel fluorescence spectrometer, Atmospheric Chem. Phys., 10(10), 4453–4466, doi:10.5194/acp-10-4453-2010, 2010.

Huffman, J. A., Treutlein, B. and Pöschl, U.: Fluorescent biological aerosol particle concentrations and size distributions measured with an Ultraviolet Aerodynamic Particle Sizer (UV-APS) in Central Europe, Atmospheric Chem. Phys., 10(7), 3215–3233, doi:10.5194/acp-10-3215-2010, 2010.

Whitehead, J. D., Gallagher, M. W., Dorsey, J. R., Robinson, N., Gabey, A. M., Coe, H., McFiggans, G., Flynn, M. J., Ryder, J., Nemitz, E. and Davies, F.: Aerosol fluxes and dynamics within and above a tropical rainforest in South-East Asia, Atmospheric Chem. Phys., 10(19), 9369–9382, doi:10.5194/acp-10-9369-2010, 2010.

Wilkinson, D. M., Koumoutsaris, S., Mitchell, E. A. D. and Bey, I.: Modelling the effect of size on the aerial dispersal of microorganisms, J. Biogeogr., 39(1), 89–97, doi:10.1111/j.1365-2699.2011.02569.x, 2012.

Zawadowicz, M. A., Froyd, K. D., Murphy, D. M. and Cziczo, D. J.: Improved identification of primary biological aerosol particlesnewline using single-particle mass spectrometry, Atmospheric Chem. Phys., 17(11), 7193–7212, doi:10.5194/acp-17-7193-2017, 2017.

**ACPD**

---

## Author Comment (AC1) · 20 Sep 2017

The comment has been uploaded as a pdf supplement for ease of reading. Moreover, since the reviewer suggested the presentation of more data, a new figure will be added to the revised paper and is, therefore, uploaded here alongside the response.

Please also note the supplement to this comment:
https://www.atmos-chem-phys-discuss.net/acp-2017-527/acp-2017-527-AC1-supplement.pdf

[Figure]

[Figure]

[Figure]

**Fig. 1.** Details on the relationship between counted CFUs by the Burkard samplers and the estimated fluxes for the campaigns of July (a) and September (b) 2015

**Supplement:**

**Authors' Response to Reviewer #1**

**All the reviewer's comments (in boldfaced red) have been numbered sequentially. After each comment the authors report their answer indicating eventual modifications that will be made to the revised version of the manuscript.**

**1) P2L22: ": : :some periods: : :some land uses." Rather vague description doesn't help the reader in getting an overview of previous work, rephrase or delete**

Following the reviewer's advice, the sentence at (P2L22) will be removed in the revised version of the paper.

**2) P2L32ff: "But while Burrows *: : :*" The whole sentence should be rephrased to improve readability. The word "species" is used but it's not entirely clear if the authors mean microbial species.**
**Also it is unclear why "in reality" emissions are different from the results from Burrows et al.. If this is a conclusion based on the submitted work, it should rather be made in the conclusions chapter.**

The sentence (P2 L32 – P3L1-2). will be removed in revised version to avoid unclear statements

**3) P3L13ff: The site description is too general (e.g. no species resolved vegetation or bare soil coverage). This is surprising given that the authors describe the existence of a large variability in emission fluxes "especially because of variation in vegetation across space (P3L3)". Was the grassland actively managed, e.g. grazed or mown during or in between sampling? Was the management comparable between different campaigns and study sites?**

The experimental field was not intensively grazed and was not actively managed during the measurement campaigns with no mowing or irrigation. This information will be added to the text (around P3 L16-17 of the old version of the manuscript).

**4) P3L24: How does the flow rate translate to Reynolds Numbers. What are the resulting losses in the sampler's intake? Are losses biased towards larger or smaller aerosols?**

The virtual impactor was designed following good design practices from (Marple and Olson, 2011). Literature data report a sampling efficiency ranging from 80 to 100 % for mildew spores (Schwarzbach, 1979) and calibration tests performed during the 2008-2010 campaign showed it to be capable of sampling aerosolized *P.syringae*. Considering its high flowrate it is operating at super-iso-mean-velocity and therefore sampling efficiency is expected to decrease for larger particles proportionally with the ratio between external wind speed and the Burkard's sampling speed (Brockmann, 2011). This information will be added to the revised version of the paper following the description of the MRG (at P4 L3 of the old uploaded version of the manuscript).

**5a-d) P3-4 Chapter 2.1: General questions regarding the employed micro-meteorological measurement technique:**

**a) Besides precise instrumentation for the concentration gradient, steady state conditions are the key restriction for applying k-theory/gradient measurements. Was the sonic data used to investigate steady state conditions, e.g. employ standard quality checks:stationarity etc. (Foken and Wichura, 1996)?**

**b) Results from detectors precision study (MRG) are visible in figure 3 only. It would be helpful to also present actual values and compare them to measured fluxes.**

**c) was scaling of measurement height by the zero-mean displacement height discarded in estimation of K due to the comparably small canopy?**

**d) Since the authors employed a fast 3d sonic anemometer, I would welcome the addition of a more data driven flux footprint and possible flow distortion evaluation besides the cited literature relationships the authors followed. Did you discard measurements from specific wind directions, e.g. situations where sensor inlets are located downwind from the tower structure?**

A more detailed description on quality checks, footprint analysis and flow distortion will be added in a new paragraph to the Supplementary Material S1. One figure will be added to the main text (Fig. 4). To summarize here:

a) the quality check (QC) flagging system from (Mauder and Foken, 2006) was employed with flags ranging from 0 (best quality) to 2 (fluxes to discard). Only 4.54% of half-hourly measurements had a flag 2 in the September 2015 campaign where EC and flux-gradient (FG) method were compared. Keeping or discarding those data didn't significantly change the convergence of the two methods, nevertheless we chose to exclude flag 2 data in the new version of the manuscript where all the analysis and figures will reflect such exclusion.

b) a new figure will be added (Figure 4) reporting the relationship between counted CFUs, fluxes and MRG. The other figures will be re-numbered accordingly. The new figure is uploaded alongside the authors' response.

c) the zero plane displacement was computed as two thirds of the average canopy height at 0.13 m, and then z was computed above such zero level. However, as the reviewer points out, the small grass height makes the computation quite insensitive to zero plane placement, but however it was considered.

d) In the EC/FG comparison experiment, grassland footprint resulted the main contributor to the measured fluxes (about 50-60% contribution). The reviewer is correct about possible flow distortion: we decided to use the Integral Turbulence Characteristics (ITC) test to assess the goodness of turbulence data, that is embedded in the QC flagging system described above. Since no flag 2 data were detected when wind was blowing through the scaffolding, we can assume the absence of significant flow distortion

**6) P4L23: "similar herbaceous species, such as*: : :*" these species were not listed for study site 1.**

The authors would kindly point out that the two major species were already indicated at P3 L16

**7) P5L7: You assume that deposition is purely driven by gravitational settling? Are other means (e.g. negative gradient between vegetation canopy and atmosphere, interception, impaction) insignificant?**

We thank Reviewer #1 and Reviewer #3 for this suggestion that allowed us to improve the modelling approach. A new version of the PLAnET model has been developed including interception and impaction effects together with gravitational settling, following a widely adopted model(Slinn, 1982). The whole model has been re-optimized and the Gompertz parameter (Eq. (2)) re-computed. The model results remain consistent with the previous version, with only small differences in the linear regressions between measured and modelled daily averages. For the 2008-2010 campaign the slope changed from 0.71 to 0.70, offset from 0.88 to 0.28 and $r^2$ from 0.59 to 0.54. For the 2015 campaign the slope changed from 1.05 to 1.31, the offset from 0.11 to -3.75 and the $r^2$ from 0.57 to 0.68, yielding an even better correlation between measured and simulated net fluxes. The relationship between the newly computed deposition velocity and friction velocity for the campaign data remains under the Gillette et al. critical threshold (Gillette et al., 1974;Gillette et al., 1997) and therefore no bias should exist in the measured fluxes. In fact, no depositional considerations are made for example by Park et al. (2011) when applying gradient method to $PM_{10}$ fluxes. The new version of the model

will be uploaded to MathWorks FileExchange and all the data and figures in the revised version of the paper will take in account deposition, impaction and interception.

As for the negative gradients, these were never observed in the current dataset. However we agree that this should require additional investigation and this *caveat* will be included in the revised version of the paper (around P10-L 21ff of the current uploaded version of the paper)

**8) P5L18: Why did you use the Lighthart and Shaffer data to express observed fluxes as a logistic function of ustar? What is the goodness of fit for m1, m2, m3 and eq.2 in general? How does the fitting errors propagate in overall model uncertainty.**

The Lighthart and Shaffer data were used to parameterize Eq. (2) independently from the validation data. Using our own measurements to obtain a fit between $u_*$ and fluxes would have probably resulted in better overall results when comparing the model to the data, but reduced the applicability of the model outside the presented situation. m1, m2, m3 are the result of an iterative numerical optimization, therefore their confidence interval could not directly be computed; we performed a sensitivity analysis to assess the impact of those coefficients on fluxes (Table 2), revealing that changing the coefficients by 10% has a small overall impact on the predicted fluxes (the delta in the error function $\varepsilon$ is < 0.5). The final adjusted $r^2$ was 0.44.

**9) P6L5: The settling velocity is highly dependent on particle diameter and shape. Obviously for the applicability of a model, simplifications have to be made here, since time resolved aerosol size and density spectra are hard to measure or predict. However it would be worth exploring if deposition (i.e. Vg) has a larger impact on overall predicted net fluxes by varying aerosol size modes and densities (in reality these will have temporal patterns for instance due to phenology of different sources). It should at least be specified if the used literature values for particle density and particle diameter are representative only for grasslands or a specific season.**

We refer to Raisi et al. (2013) that , even if related to a suburban area, is one of the most recent works were a long term monitoring of bacterial and fungal species was made and coupled with aerodynamic considerations. The seasonal and spatial variation that the reviewer's points out is absolutely true and is acknowledged as well by Raisi et al. (2013). As the reviewer correctly states, a single diameter choice was made as a simplification. In the cited work the highest fraction of cultivable fungi and bacteria was found in the range between 2.1 and 3.3 microns and we have therefore chose 3.3 μm as a cautionary representative diameter for bioaerosols in the model. The text will be modified to acknowledge the non-universality of this aerodynamic choice (around P10, L11ff of the present uploaded version of the paper).

**10) P6L10: eq. 6: please report goodness of fit for linear regression between avg C and LAI. How does the uncertainty in Ca propagate into prediction of Fd?**

Goodness of fit between the variables is rather low ($r^2$ < 0.2), however its impact on the overall model and the predicted net fluxes is limited: a 50% increase in simulated concentrations (Ca in the model) resulted in an average percentage change of net fluxes of roughly 2% on the calibration dataset.

**11) P8L5: The comparison between flux gradient and eddy covariance flux measurements provide confidence in the observations. However, eddy fluxes are not necessarily the truth, flux errors in LE are often in the magnitude of tens of percent. EC measurements were made at different height than the gradient measurements, meaning that the EC instrumentation sees different parts of the grassland. Along these lines, the open path EC sensor was not cross-calibrated with the closed path gradient sensors? In other words, It will be hard to conclude if the gradient or the EC results are off. A more detailed error/uncertainty discussion of the net fluxes obtained from the gradient method would be appreciated here. It should at least be acknowledged that the conclusions made from the H2O flux comparisons do not necessarily**

**apply to aerosol flux measurements. What are the expected uncertainties introduced through assuming scalar similarity? The MRG or precision of the detectors employed at same height could be used to propagate a flux uncertainty.**
**Since the model is calibrated on the measured fluxes (plus minus uncertainty) also the model will have this uncertainty.**

We fully agree with the reviewer in that the good correspondence between EC and FG water vapour fluxes does not necessarily apply to bioaerosol flux measurements and we will add a corresponding note of caution to the revised manuscript.

As far as the concentration profile follows M-O theory, the difference in sampling height should not impact retrieved gradient fluxes. Sampling height is an input to the method and is therefore accounted for. Our field measurement conditions overall meet M-O requirements (e.g flat terrain, homogeneous).

The reviewer is correct that flux footprints from the two methods are likely slightly different, being larger for the EC that has a higher sampling height. However, the footprint analysis we performed revealed that majority of the EC footprint is contained within the experimental field, that is very homogeneous, making the two flux measurements safely comparable. Such discussion will be added to the revised version of the supplementary material.

The comparison between EC and FG was performed in order to assess the presence of a significant bias between the two methods (i.e.: significant under or over estimation of the FG method at low/high fluxes). The latter intent will be clarified in the revised version of the paper by modifying P8 L5-6 and, in the conclusions, P12 L15-18.

**12) P8L15ff: the 2008/2009 measurements have a wider spread, partly due to the fact that no detection limit was applied (e.g. in 2015 all negative fluxes were removed due to the detection limit). What is the reason for that?**

In the 2008-2010 campaign, the samplers were calibrated with multiple replicates of aerosolized *P. syringae* at different dilutions (three dilutions at $10^2$, $10^3$ and $10^4$ bacteria ml$^{-1}$, three replicates per dilution). Given that no statistically significant differences were detected between the two samplers (except in one replicate at $10^3$ bacteria ml$^{-1}$), no MRG was computed.

This explanation will be added to the revised version of the paper alongside aerodynamic considerations about the sampler (see comment 4 above).

**13) P11L33ff: besides rainfall other events could have an effect on PBA production. You mentioned the heat wave in 2003. What about water stress or cutting/mowing/ grazing. Some of these stress factors might have lagged interactions with LAI and microorganism population growth. It would be great to introduce 1 or 2 sentences about these effects and how they would change the annual emission from a grassland, if feasible**

Following the reviewer comment, some further effects on PBAs production will be discussed in the new version of the paper (around P12 L11ff of the present uploaded version of the paper).

**14) Fig(3): Why is the detection limit (MRG) half in Sept-Oct as compared to July?**

We apologize for any lack of clarity. Figure 3 does not report MRG values, only fluxes. MRG values are calculated on the CFUs measured by the samplers and when the two samplers fall below such detection limit, the flux derived by FG method at that time is flagged (yellow star) as unreliable. The new figure 4 (see also comment 5b) should clarify the latter point

**15) Technical Corrections: P1L19: "*: : :*than that*: : :*". Numerus P3L12: "similar terrain". Phrasing: Do you mean flat? P8L7: delete ",instead," P8L7: rephrase, e.g.: spanned over multiple seasons P9L32: "*: : :* all the emission-deposition processes", please rephrase P12L14: "*: : :*has been". Numerus P12L21: "*: : :*outward fluxes". Rephrase, e.g. emission fluxes**

All the technical corrections will be addressed in the revised version of the paper where indicated

**References cited by the authors in the response:**

Brockmann, J. E.: Aerosol Transport in Sampling Lines and Inlets, in: Aerosol Measurement, John Wiley & Sons, Inc., 68-105, 2011.

Gillette, D. A., Blifford, I. H., and Fryrear, D. W.: The influence of wind velocity on the size distributions of aerosols generated by the wind erosion of soils, Journal of Geophysical Research, 79, 4068-4075, 10.1029/JC079i027p04068, 1974.

Gillette, D. A., Fryrear, D. W., Gill, T. E., Ley, T., Cahill, T. A., and Gearhart, E. A.: Relation of vertical flux of particles smaller than 10 μm to total aeolian horizontal mass flux at Owens Lake, Journal of Geophysical Research: Atmospheres, 102, 26009-26015, 10.1029/97JD02252, 1997.

Marple, V. A., and Olson, B. A.: Sampling and Measurement Using Inertial, Gravitational, Centrifugal, and Thermal Techniques, in: Aerosol Measurement, John Wiley & Sons, Inc., 129-151, 2011.

Mauder, M., and Foken, T.: Impact of post-field data processing on eddy covariance flux estimates and energy balance closure, Meteorologische Zeitschrift, 15, 597-609, 10.1127/0941-2948/2006/0167, 2006.

Park, M.-S., Park, S.-U., and Chun, Y.: Improved parameterization of dust emission (PM10) fluxes by the gradient method using the Naiman tower data at the Horqin desert in China, Science of The Total Environment, 412–413, 265-277, 10.1016/j.scitotenv.2011.09.068, 2011.

Raisi, L., Aleksandropoulou, V., Lazaridis, M., and Katsivela, E.: Size distribution of viable, cultivable, airborne microbes and their relationship to particulate matter concentrations and meteorological conditions in a Mediterranean site, Aerobiologia, 29, 233-248, 10.1007/s10453-012-9276-9, 2013.

Schwarzbach, E.: A High Throughput Jet Trap for Collecting Mildew Spores on Living Leaves, Journal of Phytopathology, 94, 165-171, 10.1111/j.1439-0434.1979.tb01546.x, 1979.

Slinn, W. G. N.: Predictions for particle deposition to vegetative canopies, Atmospheric Environment (1967), 16, 1785-1794, 10.1016/0004-6981(82)90271-2, 1982.

---

## Author Comment (AC2) · 20 Sep 2017

**Authors' Response to Reviewer #2**

**All the reviewer's comments (in boldfaced red) have been numbered sequentially. After each comment the authors report their answer indicating eventual modifications that will be made to the revised version of the manuscript.**

**1) Discussion: Please include comparison between the PLAnET estimates and the previous studies, e.g. the Burrows et al, (2009b) and microbial flux observations from other locations, for instance using the same scaling factor to total microbes as Burrows et al. (2009a) used for grasslands (302).**

We would like to thank the reviewer for this comment since it allowed us to explore potential convergences between PLAnET and ECHAM5. In fact, by looking at figure 4 in Burrows et al. (2009) the authors were able to extrapolate a median value for flux of total microorganisms for grassland (as simulated by ECHAM5) of roughly 1000 organisms $m^{-2} s^{-1}$. By scaling the PLAnET model outputs for the 2008-2010 and 2015 simulations with the factor proposed by the reviewer an average net flux of 750.49 organisms $m^{-2} s^{-1}$ was found. The latter estimate is referring to the new version of the PLAnET model including the new deposition scheme (see response to reviewer #1 and #3) and is a surprising result, considering the fact that PLAnET is still in its infancy. These comparisons will be added to the discussion of the revised version of the paper.

**2) Page 11, first paragraph suggests that obtaining a scaling factor to total microorganisms from the culturable fraction requires flux measurements of the total microorganisms. Why cannot the scaling factor be estimated from lower temporal resolution concentration measurements of both total microbes and the culturable fraction in the same site the flux measurements are made?**

It would indeed be possible assuming that the culturability of airborne microorganisms does not change during the time span of the "slower" total microorganisms samples. The latter possibility will be added to the text as a suggestion for future experiments (around P11 L11-13 of the old version of the manuscript).

**3) Page 11, lines13-18: Culturable and viable are not necessarily the same thing (see e.g Burrows et al, 2009a)**

In the new version of the paper term "viability" in P11 L5 will be changed to "culturability" and it will be clarified that epifluorescence, contrary to plate incubation, may detect viable-but-nonculturable microorganisms by adding a short sentence immediately after P11 L5.

**4) Page 2, line 20-23: "In the past, only few attempts have been made to quantify the flux of microorganisms from plant canopies (Lindemann et al., 1982; Lindemann and Upper, 1985; Lighthart and Shaffer, 1994) covering only some periods and some land uses." This is confusing, as later the authors themselves reference several other studies that have tried to quantify the fluxes using various methodology. Please specify that this sentence refers to direct measurements of bacterial fluxes.**

The suggestion of the reviewer will be implemented by changing the sentence (at P2 L20-21) "In the past, only few attempts have been made to quantify the flux of microorganisms from plant canopies" to "In the past, only few attempts have been made to directly measure the flux of bacteria".

**5) Page 4, line 23-24 "The experimental field for these previous campaigns was covered with similar herbaceous species such as cocksfoot (Dactylis glomerata), ryegrass, tall fescue (Festuca arundinacea) and alfalfa (Medicago sativa)." The list of species for the other field on the previous page only included clover and ryegrass, so how similar was the vegetation on these fields?**

We apologize for the lack of clarity. The sentence will be corrected with "Herbaceous species with similar habitus" in the new version of the paper.

**References cited by the authors in the answers**

Burrows, S. M., Butler, T., Jockel, P., Tost, H., Kerkweg, A., Poschl, U., and Lawrence, M. G.: Bacteria in the global atmosphere - Part 2: Modeling of emissions and transport between different ecosystems, Atmospheric Chemistry and Physics, 9, 9281-9297, 2009.

---

## Author Comment (AC3) · 20 Sep 2017

**Authors' Response to Reviewer #3**

**All the reviewer's comments (in boldfaced red) have been numbered sequentially. After each comment the authors report their answer indicating eventual modifications that will be made to the revised version of the manuscript.**

**1) Some equations contain errors and units are missing or incorrect for a number of parameters, see the specific comments below. There are also a number of inconsistencies between the equations in the MS and in the code. I am puzzled most by the formulation for microbial population growth: is it assumed to respond instantaneously to changes in driving variables (temperature)? If so, is that a valid assumption on the 30 min. time step that you applied here, and at which time step would this assumption brake down? Or are the dynamics of the microbial population calculated transiently?
Moreover, the formulation and units of eq. 8 are inconsistent, which is where most of my confusion comes from.**

In the model it is assumed that the microorganisms grow by a temperature-dependent factor (r) each 30 minutes (1800 s). Relatively fast growth are not unheard in field samplings with doubling times reported as low as 3.5 to 3.8 hours (see Hirano and Upper (1986) and references therein). However, fast growth in the model single time step is unlikely. Since a short time step is necessary to represent correct variation of friction velocity and other environmental parameters, the authors have introduced a calibration constant fine-tuned by the optimization procedure to adjust growth rate on such a short time-step. As for unit inconsistency please see comment 7 below.

**2) Why is only gravitational settling considered as removal mechanism? Other dry deposition mechanisms can be relevant for particles of the assumed size (3.3 um). How sensitive are the calculated dry deposition fluxes to assumptions on the particle diameter?
The title promises new insights into microbial fluxes, but I do not see them in the abstract or conclusions. What are for instance the 'underlying driving forces (P12,L25)'of microbial emissions? What new insights has the combination of the flux measurements and the emission model yielded into these driving forces? Could you highlight these findings in the abstract and conclusion?**

Please see reply to reviewer #1 comment 7 regarding the implementation of a deposition model considering impaction and interception.

About  the 3.3 microns diameter, we chose it following Raisi et al. (2013) since, even if related to a suburban area, it was one of the most recent works where a long term monitoring of bacterial and fungal species was carried out along with aerodynamic considerations. For sizes close to the chosen diameter the average deposition fluxes are quite similar to what is presented in our paper (for a doubling of the diameter, the average difference in net fluxes is < 10 % on the calibration dataset).

About the "new insights", we aimed to highlight that microbial emissions are not driven only by turbulence, but are a more complex interaction of population dynamics, surface dynamics and atmospheric conditions, as shown with the PLAnET model. We can understand, though, that these are not really "new" insights but rather a confirmation of what was already hypothesized in past works. Following the reviewer's suggestion, we will modify the title in "Measurements and modelling of surface-atmosphere exchange of microorganisms in Mediterranean grassland".

**3) I would like to see some more discussion on which types of microorganisms are sampled. The MS mentions viable microorganisms. Does that include both bacteria and fungal spores? Besides, can you say something about the size range of the observed particles? This will be important to eventually evaluate the role of the emitted bioaerosols on climate.**

The chosen sampling medium was non-selective, thus allowing the growth of both bacteria and fungal spores and this will be clarified in the new text. Burkard samplers used in this experiment have proven to be able to sample both aerosolized *P. syringae* (see reviewer #1 comment 12) and mildew spores (Schwarzbach, 1979). Reasonably, particle size ranged from few ($\approx$ 3) up to tens of microns ($\approx$ 40).

**4) P2,L21: in addition to these papers, (Crawford et al., 2014) measured PBA fluxes using the flux-gradient method, and (Ahlm et al., 2010; Whitehead et al., 2010) measured fluxes of coarse aerosol in tropical forests (presumably PBAs) using eddy-covariance**

We thank the reviewer for these further references that will be added to the revised version of the text (around P2 L22-24 of the old version of the paper).

**5) P5,L2: competition is mentioned here as a driver of the microbial dynamics, but I don't think it is actually included in the model. Please limit this description to processes that are included in the model.**

The revised version of the text will clarify that the PLAnET model represents the source term only as a temperature dependent growth function

**6) P5,L30: why is only gravitational settling included? For supermicron particles, also inertial impaction is important.**

Please see reply to comment 2 and reply to reviewer #1 comment 7 regarding the implementation of a deposition model considering impaction and interception.

**7) P6, eq8: I have some serious concerns regarding this equation, both as presented in the MS as in the code; this equation has microbial population size in the same units as the microbial growth and emission flux, which cannot be true. Should it read dN/dt=rNFn? In that case, it would represent exponential growth of the microbial population and loss due to emission. In the code, it is implemented as N(t)=N(t-1) + N(t-1)\*r + Fn, in which units are also inconsistent. It could be solved by multiplication of the 2nd and 3rd term on the RHS by the timestep, which would yield a discretization of the equation for exponential growth.**

We thank the reviewer for highlighting some inconsistency that were indeed present in the manuscript, and were corrected as detailed below. As for the model code and its numerical outputs, we understand the reviewer's concern that is due to a lack of clarity and lack of comments in the code, that however is correct and produced correct results.

Both emission and deposition fluxes are multiplied by the time step constant ($\xi$, in s, see Eq. (2) and Eq. (4)) therefore the correct dimensionality of $F_e$ and $F_d$ is CFU m$^{-2}$. New population at the time-step t is computed as the product of a growth rate (r, which is a ratio of temperatures and, therefore, non-dimensional) times the population at the time step t-1 (in CFU m$^{-2}$) times the net flux ($F_e$-$F_d$) which, thanks to the multiplication by the time step in s of $F_e$ and $F_d$, is also in CFU m$^{-2}$. Fluxes are converted back into CFU m$^{-2}$ s$^{-1}$ by dividing $F_e$, $F_d$ and $F_n$ by $\xi$ when outputting the variables, thus all the plots are consistent with the displayed units as well. We apologize for the lack of clarity in the explanation. All the explanations made here will be added to the revised version the paper immediately following Eq.(2) and (4). More comments will also be added to the revised version of the model that will be uploaded to MathWorks FileExchange.

**8) P6,L24: it is unclear what is meant here: 'kmin, which is the point at which all process find an equilibrium'**

Sentence will be rephrased in the revised version of the paper to better understand the link with the reference of Waggoner (1973) (i.e.: the presence of a fraction of microorganisms sheltered by wind action).

**9) P7, L14: can you discuss how this choice has affected your results? This number seems to be important in determining the upper and lower bounds of the modeled microbial population.**

The choice was made to avoid the unrealistic scenario of having always all the plants' leaves exposed to the atmosphere. However, it is a simple scaling of parameters ranging 1-2 orders of magnitude and, as it is possible to see from the model sensitivity analysis (Table 2), 10 % variations in $k_{min}$ and $k_{max}$ have minimal impact on model performance (values of $\varepsilon < 0.24$).

**10) P9,L28-31: strictly spoken, the Burrows et al 2009a study does not discuss the effect of PBAP on precipitation, which is what this sentence seems to imply**

Following the reviewer's advice, we changed the sentence from "Previous attempts to understand the impact of PBAs on precipitation" to "Previous attempts to understand the distribution of PBAs in the atmosphere".

**11) P9,L32: I would add transport to 'emission-deposition process' (e.g. Wilkinson et al., 2012)**

Following the reviewer's advice, we will add such process and reference at the indicated point in the text

**12) P10,L28: what does it mean if the 95% confidence intervals include 0 and 1 or not?**

The authors referred to the confidence interval for the slope of the regression. The fact that such intervals do not contain 0 implies that, even taking in account the uncertainties, the slope remains significantly different from zero (i.e. >0). Since the confidence intervals cross 1 we cannot exclude that by increasing the data point a better linear regression (with a slope close to the 1:1 line) could be achieved.

**13) P11,L2-12: I miss a discussion here on the use of online detection of PBAs using fluorescence measurements (e.g. Gabey et al., 2010; Huffman et al., 2010) or single particle mass-spectrometry (Zawadowicz et al., 2017). These techniques measure concentrations, but could in principle be used in combination with micrometeorological techniques to measure fluxes (e.g. Crawford et al., 2014).**

A brief discussion will be added in the revised version of the paper addressing the possibility of applying UV-LIF and SPMS to measuring microbial emissions.

**14) P11,L18: it is unclear what is meant here: 'it is not to underestimate the long-term importance of evaluating the viable fraction of said fluxes'. Please rephrase**

The sentence implied that, from the evolutionary point of view as well as for any interest in pathogen transport and colonization of distance places it is important to understand how many viable microorganisms are leaving the surface, since that would be the fraction of the total microorganisms potentially able to reproduce, colonize and, eventually, attack new areas and hosts. Nevertheless we agree with the reviewer that the significance of the sentence was not really explicit. In the new revised version of the paper the sentence will be rephrased.

**15) P12,L9-10: is rain rate given in mm/hour here?**

That is correct. The unit will be added to the revised version of the paper.

**16) Fig. 6: with half-hourly observations and model data available, why are only daily average fluxes given? In addition, it would be interesting to see time series of observations and model data.**

The model needed to run at half-hourly time steps for resolving meteorological dynamics, but then daily averages were calculated in order to reduce the significant random uncertainty inherent to the 30 minutes' data.

**Technical issues**
**17) P4,L15: unit is missing for z0**

Unit (m) will be added to the revised version of the paper.

**18) P5, eq 2: in the code, Nk_max is given as N/k_max, which seems correct to me, as it would express the population scaled by the carrying capacity, and judging by the units. Besides, the values of m1-m3 differ slightly from those in L19. What are the units of m1-m3? They cannot all be unitless (as mentioned in Table 1 and 2) when Fe is in CFU m-2 s-1.**

Thank you for pointing this out. Units for m1 and m2 are in CFU $m^{-2}$ $s^{-1}$, while m3 is adimensional. Table 1 will be updated in the new version of the paper.

**19) P6, eq 9: this equation seems to be missing an exponent ((Topt-Tmin)/(Tmax-Topt)), which is included in the code. What is the unit of r? Based on eq. 8 it should be s-1. Then also c should have this unit, and not none, as mentioned in Table 1 and 2. Please check these and other units throughout the MS.**

We thank the reviewer for spotting the error, the exponent for Eq. (9) will be added to the new version of the text. As for the unit of r it is the result of a ratio of temperatures and an adimensional calibration constant (c, see Eq.(9)) and is therefore adimensional. See also the answer to comment 7.

**20) Miscellaneous typos and language corrections:**
**P9,L9: won't -> will not**
**P10,L23: remove 'it'**
**P10,L24: the Planet -> Planet**
**P11,L1: a scaling -> scaling**
**P11,L14: transmit -> transmitting**
**P11,L15: represents -> represent**
**P11,L16: the atmospheric -> atmospheric**
**P12,L6: acting -> act**

**P12, L32: which is nested -> which it is nested**
**P12,L14: has-> have**
**P12,L24: suggest adding a comma between 'precipitation and'**
**P12,L32: which is -> which it is**

Technical adjustments were made following the reviewer's suggestions.

**21) Fig. 3 and 5: Data within years are plotted as if they represent time series (with continuous lines), but this is not always the case. This makes the plots hard to interpret.
Besides, time labels are placed at irregular intervals. Pls update these figures to make them easier to understand.**

In the new version of the paper the figures will be reworked in order to remove lines, have clearer time labels and overall give a better presentation of the flux measurements.

**22) In the code at L305: in the Cc calculation, a factor of 2 is missing in the exponent**

We thank the reviewer for pointing out the mistake. The correction was applied, but the presented results are not heavily influenced by said mistake. With the introduction of the new impaction/interception model the settling velocity alone has become almost negligible. This correction will nevertheless be applied in the new version of the PLAnET model that will be uploaded to the MathWorks File Exchange.

**References cited by the authors in the answers**

Hirano, S., and Upper, C.: Temporal, spatial, and genetic variability of leaf-associated bacterial populations, Microbiology of the phyllosphere/edited by NJ Fokkema and J. van den Heuvel, 1986.

Schwarzbach, E.: A High Throughput Jet Trap for Collecting Mildew Spores on Living Leaves, Journal of Phytopathology, 94, 165-171, 10.1111/j.1439-0434.1979.tb01546.x, 1979.

Waggoner, P.: removal of Helminthosporium maydis spores by wind, Phytopathology, 1973.

---

## Author Response (AR1)

**Point-to-point reviewers' response:**

**All the reviewer's comments (in boldfaced red) have been numbered sequentially. After each comment the authors report their answer indicating eventual modifications made in the revised version of the manuscript. Indicated line numbers are relative to the track change version of the manuscript (called "revised manuscript" from here onward) in order to ease the browsing of the manuscript itself and assess changes to the original text. Also, in the track change version of the manuscript, main changes have been highlighted with a comment in the form of "Rn-Cn" where Rn indicates the reviewer number (1-3) and Cn the comment number.**

**REVIEWER #1**

**1) P2L22: ": : :some periods: : :some land uses." Rather vague description doesn't help the reader in getting an overview of previous work, rephrase or delete**

Following the reviewer's advice, the sentence at **P2 L23** of the revised manuscript has been removed.

**2) P2L32ff: "But while Burrows *: : :*" The whole sentence should be rephrased to improve readability. The word "species" is used but it's not entirely clear if the authors mean microbial species.**
**Also it is unclear why "in reality" emissions are different from the results from Burrows et al.. If this is a conclusion based on the submitted work, it should rather be made in the conclusions chapter.**

The sentence at **P3 L2 – 4** of the revised manuscript has been removed to avoid unclear statements

**3) P3L13ff: The site description is too general (e.g. no species resolved vegetation or bare soil coverage). This is surprising given that the authors describe the existence of a large variability in emission fluxes "especially because of variation in vegetation across space (P3L3)". Was the grassland actively managed, e.g. grazed or mown during or in between sampling? Was the management comparable between different campaigns and study sites?**

The experimental field was not intensively grazed and was not actively managed during the measurement campaigns with no mowing or irrigation. This information has been added at **P3 L18-19** of the revised manuscript.

**4) P3L24: How does the flow rate translate to Reynolds Numbers. What are the resulting losses in the sampler's intake? Are losses biased towards larger or smaller aerosols?**

The virtual impactor was designed following good design practices from (Marple and Olson, 2011). Literature data report a sampling efficiency ranging from 80 to 100 % for mildew spores (Schwarzbach, 1979) and calibration tests performed during the 2008-2010 campaign showed it to be capable of sampling aerosolized *P.syringae*. Considering its high flowrate it is operating at super-iso-mean-velocity and therefore sampling efficiency is expected to decrease for larger particles proportionally with the ratio between external wind speed and the Burkard's sampling speed (Brockmann, 2011). This information has been added to **P4 L3-8** of the revised manuscript.

**5a-d) P3-4 Chapter 2.1: General questions regarding the employed micro-meteorological measurement technique:**

**a) Besides precise instrumentation for the concentration gradient, steady state conditions are the key restriction for applying k-theory/gradient measurements. Was the sonic data used to investigate steady state conditions, e.g. employ standard quality checks:stationarity etc. (Foken and Wichura, 1996)?**

**b) Results from detectors precision study (MRG) are visible in figure 3 only. It would be helpful to also present actual values and compare them to measured fluxes.**

**c) was scaling of measurement height by the zero-mean displacement height discarded in estimation of K due to the comparably small canopy?**

**d) Since the authors employed a fast 3d sonic anemometer, I would welcome the addition of a more data driven flux footprint and possible flow distortion evaluation besides the cited literature relationships the authors followed. Did you discard measurements from specific wind directions, e.g. situations where sensor inlets are located downwind from the tower structure?**

A more detailed description on quality checks, footprint analysis and flow distortion has been added in a new paragraph to the Supplementary Material S1. **One figure has been added to the main text (Fig. 4)**. To summarize here:

a) the quality check (QC) flagging system from (Mauder and Foken, 2006) was employed with flags ranging from 0 (best quality) to 2 (fluxes to discard). Only 4.54% of half-hourly measurements had a flag 2 in the September 2015 campaign where EC and flux-gradient (FG) method were compared. Keeping or discarding those data didn't significantly change the convergence of the two methods, nevertheless we chose to exclude flag 2 data in the new version of the manuscript where all the analysis and figures now reflect such exclusion.

b) a new figure will be added (Figure 4) reporting the relationship between counted CFUs, fluxes and MRG. The other figures will be re-numbered accordingly. The new figure has been inserted in the revised manuscript. The figure is introduced in the text at **P8 L5-6**

c) the zero plane displacement was computed as two thirds of the average canopy height at 0.13 m, and then z was computed above such zero level. However, as the reviewer points out, the small grass height makes the computation quite insensitive to zero plane placement, but however it was considered.

d) In the EC/FG comparison experiment, grassland footprint resulted the main contributor to the measured fluxes (about 50-60% contribution). The reviewer is correct about possible flow distortion: we decided to use the Integral Turbulence Characteristics (ITC) test to assess the goodness of turbulence data, that is embedded in the QC flagging system described above. Since no flag 2 data were detected when wind was blowing through the scaffolding, we can assume the absence of significant flow distortion

**6) P4L23: "similar herbaceous species, such as*: : :*" these species were not listed for study site 1.**

The authors would kindly point out that the two major species were already indicated at P3 L16 of the original manuscript (now **P3 L18**).

**7) P5L7: You assume that deposition is purely driven by gravitational settling? Are other means (e.g. negative gradient between vegetation canopy and atmosphere, interception, impaction) insignificant?**

We thank Reviewer #1 and Reviewer #3 for this suggestion that allowed us to improve the modelling approach. A new version of the PLAnET model has been developed including interception and impaction effects together with gravitational settling, following a widely adopted model(Slinn, 1982). The whole model has been re-optimized and the Gompertz parameter (Eq. (2)) re-computed. The model results remain consistent with the previous version, with only small differences in the linear regressions between measured and modelled daily averages. For the 2008-2010 campaign the slope changed from 0.71 to 0.70, offset from 0.88 to 0.28 and $r^2$ from 0.59 to 0.54. For the 2015 campaign the slope changed from 1.05 to 1.31, the offset from 0.11 to -3.75 and the $r^2$ from 0.57 to 0.68,

yielding an even better correlation between measured and simulated net fluxes. The relationship between the newly computed deposition velocity and friction velocity for the campaign data remains under the Gillette et al. critical threshold (Gillette et al., 1974;Gillette et al., 1997) and therefore no bias should exist in the measured fluxes. In fact, no depositional considerations are made for example by Park et al. (2011) when applying gradient method to $PM_{10}$ fluxes. The new version of the model will be uploaded to MathWorks FileExchange and all the data and figures in the revised version of the paper will take in account deposition, impaction and interception. The revised version of the paper acknowledge the insertion of the new deposition model and its equations at **P5 L 23-27, P6 L 13-17, P6 L 23-30, P7 L1-6**

As for the negative gradients, these were never observed in the current dataset. However we agree that this should require additional investigation and this *caveat* has been included in the revised version of the paper (**P13 L 23-26**)

**8) P5L18: Why did you use the Lighthart and Shaffer data to express observed fluxes as a logistic function of ustar? What is the goodness of fit for m1, m2, m3 and eq.2 in general? How does the fitting errors propagate in overall model uncertainty.**

The Lighthart and Shaffer data were used to parameterize Eq. (2) independently from the validation data. Using our own measurements to obtain a fit between $u_*$ and fluxes would have probably resulted in better overall results when comparing the model to the data, but reduced the applicability of the model outside the presented situation. m1, m2, m3 are the result of an iterative numerical optimization (this clarification has been added at **P6 L6**), therefore their confidence interval could not directly be computed; we performed a sensitivity analysis to assess the impact of those coefficients on fluxes (Table 2), revealing that changing the coefficients by 10% has a small overall impact on the predicted fluxes (the delta in the error function $\varepsilon$ is < 0.5). The final adjusted $r^2$ was 0.44.

**9) P6L5: The settling velocity is highly dependent on particle diameter and shape. Obviously for the applicability of a model, simplifications have to be made here, since time resolved aerosol size and density spectra are hard to measure or predict. However it would be worth exploring if deposition (i.e. Vg) has a larger impact on overall predicted net fluxes by varying aerosol size modes and densities (in reality these will have temporal patterns for instance due to phenology of different sources). It should at least be specified if the used literature values for particle density and particle diameter are representative only for grasslands or a specific season.**

We refer to Raisi et al. (2013) that , even if related to a suburban area, is one of the most recent works were a long term monitoring of bacterial and fungal species was made and coupled with aerodynamic considerations. The seasonal and spatial variation that the reviewer's points out is absolutely true and is acknowledged as well by Raisi et al. (2013). As the reviewer correctly states, a single diameter choice was made as a simplification. In the cited work the highest fraction of cultivable fungi and bacteria was found in the range between 2.1 and 3.3 microns and we have therefore chose 3.3 μm as a cautionary representative diameter for bioaerosols in the model. The text has been modified to acknowledge the non-universality of this aerodynamic choice (**P11 – L18-21**).

**10) P6L10: eq. 6: please report goodness of fit for linear regression between avg C and LAI. How does the uncertainty in Ca propagate into prediction of Fd?**

Goodness of fit between the variables is rather low ($r^2 < 0.2$), however its impact on the overall model and the predicted net fluxes is limited: a 50% increase in simulated concentrations (Ca in the model) resulted in an average percentage change of net fluxes of roughly 2% on the calibration dataset.

**11) P8L5: The comparison between flux gradient and eddy covariance flux measurements provide confidence in the observations. However, eddy fluxes are not necessarily the truth, flux errors in LE are often in the magnitude of tens of percent. EC measurements were made at**

**different height than the gradient measurements, meaning that the EC instrumentation sees different parts of the grassland. Along these lines, the open path EC sensor was not cross-calibrated with the closed path gradient sensors? In other words, It will be hard to conclude if the gradient or the EC results are off. A more detailed error/uncertainty discussion of the net fluxes obtained from the gradient method would be appreciated here. It should at least be acknowledged that the conclusions made from the H2O flux comparisons do not necessarily apply to aerosol flux measurements. What are the expected uncertainties introduced through assuming scalar similarity? The MRG or precision of the detectors employed at same height could be used to propagate a flux uncertainty.**
**Since the model is calibrated on the measured fluxes (plus minus uncertainty) also the model will have this uncertainty.**

We fully agree with the reviewer in that the good correspondence between EC and FG water vapour fluxes does not necessarily apply to bioaerosol flux measurements and we will add a corresponding note of caution to the revised manuscript (**P14 L30-32, P15 L1**)

As far as the concentration profile follows M-O theory, the difference in sampling height should not impact retrieved gradient fluxes. Sampling height is an input to the method and is therefore accounted for. Our field measurement conditions overall meet M-O requirements (e.g flat terrain, homogeneous).

The reviewer is correct that flux footprints from the two methods are likely slightly different, being larger for the EC that has a higher sampling height. However, the footprint analysis we performed revealed that majority of the EC footprint is contained within the experimental field, that is very homogeneous, making the two flux measurements safely comparable. Such discussion will be added to the revised version of the supplementary material.

The comparison between EC and FG was performed in order to assess the presence of a significant bias between the two methods (i.e.: significant under or over estimation of the FG method at low/high fluxes). The latter intent has been clarified in the revised version of the paper by modifying **P9 L8-9** and, in the conclusions, **P14 L30-32** and **P15 L1**.

**12) P8L15ff: the 2008/2009 measurements have a wider spread, partly due to the fact that no detection limit was applied (e.g. in 2015 all negative fluxes were removed due to the detection limit). What is the reason for that?**

In the 2008-2010 campaign, the samplers were calibrated with multiple replicates of aerosolized *P. syringae* at different dilutions (three dilutions at $10^2$, $10^3$ and $10^4$ bacteria ml$^{-1}$, three replicates per dilution). Given that no statistically significant differences were detected between the two samplers (except in one replicate at $10^3$ bacteria ml$^{-1}$), no MRG was computed.

This explanation has been added to the revised version of the manuscript (**P5 L2-6**).

**13) P11L33ff: besides rainfall other events could have an effect on PBA production. You mentioned the heat wave in 2003. What about water stress or cutting/mowing/ grazing. Some of these stress factors might have lagged interactions with LAI and microorganism population growth. It would be great to introduce 1 or 2 sentences about these effects and how they would change the annual emission from a grassland, if feasible**

Following the reviewer comment, some further effects on PBAs production have been discussed in the revised version of the paper (**P14 L18-26**).

**14) Fig(3): Why is the detection limit (MRG) half in Sept-Oct as compared to July?**

We apologize for any lack of clarity. Figure 3 does not report MRG values, only fluxes. MRG values are calculated on the CFUs measured by the samplers and when the two samplers fall below such detection limit, the flux derived by FG method at that time is flagged (yellow star) as unreliable. The new figure 4 (see also comment 5b) should clarify the latter point

**15) Technical Corrections: P1L19: ":  :  :than that:  :  :". Numerus P3L12: "similar terrain". Phrasing: Do you mean flat? P8L7: delete ",instead," P8L7: rephrase, e.g.: spanned over multiple seasons P9L32: ":  :  : all the emission-deposition processes", please rephrase P12L14: ":  :  :has been". Numerus P12L21: ":  :  :outward fluxes". Rephrase, e.g. emission fluxes**

All the technical corrections have been addressed in the revised version of the paper where indicated

**REVIEWER #2**

**1) Discussion: Please include comparison between the PLAnET estimates and the previous studies, e.g. the Burrows et al, (2009b) and microbial flux observations from other locations, for instance using the same scaling factor to total microbes as Burrows et al. (2009a) used for grasslands (302).**

We would like to thank the reviewer for this comment since it allowed us to explore potential convergences between PLAnET and ECHAM5. In fact, by looking at figure 4 in Burrows et al. (2009) the authors were able to extrapolate a median value for flux of total microorganisms for grassland (as simulated by ECHAM5) of roughly 1000 organisms $m^{-2} s^{-1}$. By scaling the PLAnET model outputs for the 2008-2010 and 2015 simulations with the factor proposed by the reviewer an average net flux of 750.49 organisms $m^{-2} s^{-1}$ was found. The latter estimate is referring to the new version of the PLAnET model including the new deposition scheme (see response to reviewer #1 and #3) and is a surprising result, considering the fact that PLAnET is still in its infancy. These comparisons, along with the relative considerations, have been added to the revised version of the paper (**P12 L15-22**).

**2) Page 11, first paragraph suggests that obtaining a scaling factor to total microorganisms from the culturable fraction requires flux measurements of the total microorganisms. Why cannot the scaling factor be estimated from lower temporal resolution concentration measurements of both total microbes and the culturable fraction in the same site the flux measurements are made?**

It would indeed be possible assuming that the culturability of airborne microorganisms does not change during the time span of the "slower" total microorganisms samples. The latter possibility has been added to the text as a suggestion for future experiments (**P12 L33-34** and **P13 L1-2**).

**3) Page 11, lines13-18: Culturable and viable are not necessarily the same thing (see e.g Burrows et al, 2009a)**

In the revised version of the paper term "viability" in **P12 L25** has been changed to "culturability" and it has been clarified that epifluorescence, contrary to plate incubation, may detect viable-but-nonculturable microorganisms by adding a short sentence immediately after (**P12 L25-27**).

**4) Page 2, line 20-23: "In the past, only few attempts have been made to quantify the flux of microorganisms from plant canopies (Lindemann et al., 1982; Lindemann and Upper, 1985; Lighthart and Shaffer, 1994) covering only some periods and some land uses." This is confusing, as later the authors themselves reference several other studies that have tried to quantify the fluxes using various methodology. Please specify that this sentence refers to direct measurements of bacterial fluxes.**

The suggestion of the reviewer has been implemented by changing the sentence "In the past, only few attempts have been made to quantify the flux of microorganisms from plant canopies" to "In the past,

only few attempts have been made to directly measure the flux of bacteria" (**P2 L20-21** in the revised version).

**5) Page 4, line 23-24 "The experimental field for these previous campaigns was covered with similar herbaceous species such as cocksfoot (Dactylis glomerata), ryegrass, tall fescue (Festuca arundinacea) and alfalfa (Medicago sativa)." The list of species for the other field on the previous page only included clover and ryegrass, so how similar was the vegetation on these fields?**

We apologize for the lack of clarity. The sentence has been corrected with "Herbaceous species with similar habitus" in the revised version of the paper (**P4 L34**).

**REVIEWER #3**

**1) Some equations contain errors and units are missing or incorrect for a number of parameters, see the specific comments below. There are also a number of inconsistencies between the equations in the MS and in the code. I am puzzled most by the formulation for microbial population growth: is it assumed to respond instantaneously to changes in driving variables (temperature)? If so, is that a valid assumption on the 30 min. time step that you applied here, and at which time step would this assumption brake down? Or are the dynamics of the microbial population calculated transiently?**
**Moreover, the formulation and units of eq. 8 are inconsistent, which is where most of my confusion comes from.**

In the model it is assumed that the microorganisms grow by a temperature-dependent factor (r) each 30 minutes (1800 s). Relatively fast growth are not unheard in field samplings with doubling times reported as low as 3.5 to 3.8 hours (see Hirano and Upper (1986) and references therein). However, fast growth in the model single time step is unlikely. Since a short time step is necessary to represent correct variation of friction velocity and other environmental parameters, the authors have introduced a calibration constant fine-tuned by the optimization procedure to adjust growth rate on such a short time-step. As for unit inconsistency please see comment 7 below.

**2) Why is only gravitational settling considered as removal mechanism? Other dry deposition mechanisms can be relevant for particles of the assumed size (3.3 um). How sensitive are the calculated dry deposition fluxes to assumptions on the particle diameter? The title promises new insights into microbial fluxes, but I do not see them in the abstract or conclusions. What are for instance the 'underlying driving forces (P12,L25)'of microbial emissions? What new insights has the combination of the flux measurements and the emission model yielded into these driving forces? Could you highlight these findings in the abstract and conclusion?**

Please see reply to reviewer #1 comment 7 regarding the implementation of a deposition model considering impaction and interception.

About the 3.3 microns diameter, we chose it following Raisi et al. (2013) since, even if related to a suburban area, it was one of the most recent works where a long term monitoring of bacterial and fungal species was carried out along with aerodynamic considerations. For sizes close to the chosen diameter the average deposition fluxes are quite similar to what is presented in our paper (for a doubling of the diameter, the average difference in net fluxes is < 10 % on the calibration dataset).

About the "new insights", we aimed to highlight that microbial emissions are not driven only by turbulence, but are a more complex interaction of population dynamics, surface dynamics and atmospheric conditions, as shown with the PLAnET model. We can understand, though, that these are

not really "new" insights but rather a confirmation of what was already hypothesized in past works. Following the reviewer's suggestion, title has been modified in "Measurements and modelling of surface-atmosphere exchange of microorganisms in Mediterranean grassland" (**P1 L1-5**).

**3) I would like to see some more discussion on which types of microorganisms are sampled. The MS mentions viable microorganisms. Does that include both bacteria and fungal spores? Besides, can you say something about the size range of the observed particles? This will be important to eventually evaluate the role of the emitted bioaerosols on climate.**

The chosen sampling medium was non-selective, thus allowing the growth of both bacteria and fungal spores and this will be clarified in the new text. Burkard samplers used in this experiment have proven to be able to sample both aerosolized *P. syringae* (see reviewer #1 comment 12) and mildew spores (Schwarzbach, 1979). Reasonably, particle size ranged from few ($\approx 3$) up to tens of microns ($\approx 40$).

**4) P2,L21: in addition to these papers, (Crawford et al., 2014) measured PBA fluxes using the flux-gradient method, and (Ahlm et al., 2010; Whitehead et al., 2010) measured fluxes of coarse aerosol in tropical forests (presumably PBAs) using eddy-covariance**

We thank the reviewer for these further references that have been added to the revised version of the text (**P2 L22-25**).

**5) P5,L2: competition is mentioned here as a driver of the microbial dynamics, but I don't think it is actually included in the model. Please limit this description to processes that are included in the model.**

The revised version of the text clarifies that the PLAnET model represents the source term only as a temperature dependent growth function (**P5 L19-20**).

**6) P5,L30: why is only gravitational settling included? For supermicron particles, also inertial impaction is important.**

Please see reply to comment 2 and reply to reviewer #1 comment 7 regarding the implementation of a deposition model considering impaction and interception.

**7) P6, eq8: I have some serious concerns regarding this equation, both as presented in the MS as in the code; this equation has microbial population size in the same units as the microbial growth and emission flux, which cannot be true. Should it read dN/dt=rNFn? In that case, it would represent exponential growth of the microbial population and loss due to emission. In the code, it is implemented as N(t)=N(t-1) + N(t-1)\*r + Fn, in which units are also inconsistent. It could be solved by multiplication of the 2nd and 3rd term on the RHS by the timestep, which would yield a discretization of the equation for exponential growth.**

We thank the reviewer for highlighting some inconsistency that were indeed present in the manuscript, and were corrected as detailed below. As for the model code and its numerical outputs, we understand the reviewer's concern that is due to a lack of clarity and lack of comments in the code, that however is correct and produced correct results.

Both emission and deposition fluxes are multiplied by the time step constant ($\xi$, in s, see Eq. (2) and Eq. (4)) therefore the correct dimensionality of $F_e$ and $F_d$ is CFU m$^{-2}$. New population at the time-step

t (**Eq. (10)** of the revised version) is computed as the product of a growth rate (r, which is a ratio of temperatures and, therefore, non-dimensional) times the population at the time step t-1 (in CFU $m^{-2}$) times the net flux ($F_e$-$F_d$) which, thanks to the multiplication by the time step in s of $F_e$ and $F_d$, is also in CFU $m^{-2}$. Fluxes are converted back into CFU $m^{-2}$ $s^{-1}$ by dividing $F_e$, $F_d$ and $F_n$ by $\xi$ when outputting the variables, thus all the plots are consistent with the displayed units as well.

To make this process more clear in the new version of the paper the time-step multiplication has been removed from equation 2 and 4 (yielding flux in CFU $m^{-2}$ $s^{-1}$) and added to the net flux term of equation 10 (thus correctly subtracting CFU $m^{-2}$ from the population in CFU $m^{-2}$) (**P7 L23-25**). This streamlines the explanation of the model processes without affecting the results. Corrections have been made to Table 1 accordingly.

**8) P6,L24: it is unclear what is meant here: 'kmin, which is the point at which all process find an equilibrium'**

The sentence (**P7 L26-27**) has been rephrased in the revised version of the paper to better understand the link with the reference of Waggoner (1973) (i.e.: the presence of a fraction of microorganisms sheltered by wind action).

**9) P7, L14: can you discuss how this choice has affected your results? This number seems to be important in determining the upper and lower bounds of the modeled microbial population.**

The choice was made to avoid the unrealistic scenario of having always all the plants' leaves exposed to the atmosphere. However, it is a simple scaling of parameters ranging 1-2 orders of magnitude and, as it is possible to see from the model sensitivity analysis (Table 2), 10 % variations in $k_{min}$ and $k_{max}$ have minimal impact on model performance (values of $\varepsilon < 0.24$).

**10) P9,L28-31: strictly spoken, the Burrows et al 2009a study does not discuss the effect of PBAP on precipitation, which is what this sentence seems to imply**

Following the reviewer's advice, we changed the sentence from "Previous attempts to understand the impact of PBAs on precipitation" to "Previous attempts to understand the distribution of PBAs in the atmosphere" (**P10 L32-33**).

**11) P9,L32: I would add transport to 'emission-deposition process' (e.g. Wilkinson et al., 2012)**

Following the reviewer's advice, we added such processes and reference at the indicated point in the text (**P11 L3-4** in the revised version of the paper).

**12) P10,L28: what does it mean if the 95% confidence intervals include 0 and 1 or not?**

The authors referred to the confidence interval for the slope of the regression. The fact that such intervals do not contain 0 implies that, even taking in account the uncertainties, the slope remains significantly different from zero (i.e. >0). Since the confidence intervals cross 1 we cannot exclude that by increasing the data point a better linear regression (with a slope close to the 1:1 line) could be achieved.

**13) P11,L2-12: I miss a discussion here on the use of online detection of PBAs using fluorescence measurements (e.g. Gabey et al., 2010; Huffman et al., 2010) or single particle mass-spectrometry (Zawadowicz et al., 2017). These techniques measure concentrations, but could in principle be used in combination with micrometeorological techniques to measure fluxes (e.g. Crawford et al., 2014).**

A brief discussion has been added in the revised version of the paper addressing the possibility of applying UV-LIF and SPMS to measuring microbial emissions (**P12 L33-34** and **P13 L1-13**).

**14) P11,L18: it is unclear what is meant here: 'it is not to underestimate the long-term importance of evaluating the viable fraction of said fluxes'. Please rephrase**

The sentence implied that, from the evolutionary point of view as well as for any interest in pathogen transport and colonization of distance places it is important to understand how many viable microorganisms are leaving the surface, since that would be the fraction of the total microorganisms potentially able to reproduce, colonize and, eventually, attack new areas and hosts. Nevertheless we agree with the reviewer that the significance of the sentence was not really explicit. In the new revised version of the paper the sentence has been rephrased (**P13 L19-21**).

**15) P12,L9-10: is rain rate given in mm/hour here?**

That is correct. The unit has been added to the revised version of the paper (**P14 L15-16**).

**16) Fig. 6: with half-hourly observations and model data available, why are only daily average fluxes given? In addition, it would be interesting to see time series of observations and model data.**

The model needed to run at half-hourly time steps for resolving meteorological dynamics, but then daily averages were calculated in order to reduce the significant random uncertainty inherent to the 30 minutes' data.

**Technical issues**
**17) P4,L15: unit is missing for z0**

Unit (m) has been added to the revised version of the paper (**P4 L26**)

**18) P5, eq 2: in the code, Nk_max is given as N/k_max, which seems correct to me, as it would express the population scaled by the carrying capacity, and judging by the units. Besides, the values of m1-m3 differ slightly from those in L19. What are the units of m1-m3? They cannot all be unitless (as mentioned in Table 1 and 2) when Fe is in CFU m-2 s-1.**

Thank you for pointing this out. Units for m1 and m2 are in CFU $m^{-2}$ $s^{-1}$, while m3 is adimensional. Table 1 has been updated in the new version of the paper.

**19) P6, eq 9: this equation seems to be missing an exponent ((Topt-Tmin)/(Tmax-Topt)), which is included in the code. What is the unit of r? Based on eq. 8 it should be s-1. Then also c should have this unit, and not none, as mentioned in Table 1 and 2. Please check these and other units throughout the MS.**

We thank the reviewer for spotting the error, the exponent for Eq. (9) (**Eq. (11)** in the revised version of the paper) has been added to the new version of the text. As for the unit of r it is the result of a ratio of temperatures and an adimensional calibration constant (c, see **Eq.(11)** of the revised version of the paper) and is therefore adimensional. See also the answer to comment 7.

**20) Miscellaneous typos and language corrections:**
**P9,L9: won't -> will not**
**P10,L23: remove 'it'**
**P10,L24: the Planet -> Planet**
**P11,L1: a scaling -> scaling**
**P11,L14: transmit -> transmitting**
**P11,L15: represents -> represent**
**P11,L16: the atmospheric -> atmospheric**
**P12,L6: acting -> act**
**P12, L32: which is nested -> which it is nested**
**P12,L14: has-> have**
**P12,L24: suggest adding a comma between 'precipitation and'**
**P12,L32: which is -> which it is**

Technical adjustments were made following the reviewer's suggestions.

**21) Fig. 3 and 5: Data within years are plotted as if they represent time series (with continuous lines), but this is not always the case. This makes the plots hard to interpret.**
**Besides, time labels are placed at irregular intervals. Pls update these figures to make them easier to understand.**

In the new version of the paper the figures have been reworked in order to remove lines, have clearer time labels and overall give a better presentation of the flux measurements (**Figures 3 and 6**).

**22) In the code at L305: in the Cc calculation, a factor of 2 is missing in the exponent**

We thank the reviewer for pointing out the mistake. The correction was applied, but the presented results are not heavily influenced by said mistake. With the introduction of the new impaction/interception model the settling velocity alone has become almost negligible. This correction has been nevertheless applied in the new version of the PLAnET model that has been uploaded to the MathWorks File Exchange.

**References cited in the point-to-point response**

[revised manuscript text omitted]

The actual height z, considered in the aforementioned computations should actually be corrected by subtracting the zero plane displacement height, roughly two-thirds of the average canopy height.

In the present work, due to the considerably short and simple canopy, z is quite insensitive to such correction.

**2 Eddy- Covariance Quality Check and Relationship with Flux Gradient results**

The eddy-covariance (EC) technique was used to evaluate the applicability of flux-gradient (FG) technique for the Montfavet situation during the September 2015 campaign.

EC data were elaborated with EddyPro 6 and quality-checked following Mauder and Foken (2006). The flagging methodology followed the CarboEurope IP project standards with a flag of 0 indicating best quality fluxes, a flag of 1 fluxes suitable for general analysis and a flag of 2 fluxes to be discarded. In the September campaign where EC and FG were compared 84.85% of fluxes used for comparison obtained a flag of 0, 10.6% a flag of 1 and only 4.54% a flag of 2. The linear regression between EC and FG did not significantly change when removing flag 2 fluxes. The correlation coefficient changed from 0.72 to 0.70 and slope and offset where substantially unchanged (a difference of 0.005 for the slope and of 0.019 for the offset).

A two dimensional footprint model from Kljun et al. (2015) was used to investigate the contribution of the grassland to the measured flux. One aggregated footprint per day was generated, each one spanning a domain of a 25*z meter radius around the tower (76 m) with a resolution of 2 m on both the x and y direction.

Footprint analysis showed that grassland is the main contributor to the footprint with a cumulative 50-60% contribution. Fluxes are also influenced by terrain with various roughness elements that were taken into account by choosing a conservative z0 of 0.15 m.

Footprint analysis also allows investigating potential flow distortions due to the presence of the scaffolding by comparing wind direction and quality flags. During the days between the 26th and 30th of September 2015, the main wind direction was blowing through the scaffolding before reaching the sensors. Between the 26th and the 28th of September, no matter the mainly northerly wind direction, all data had a quality flag of 0, which, from the point of view of the Integral Turbulence Test, means a deviation of less than 30% compared to what is predicted based on similarity theory. Larger errors, with discrepancies > 100%, happened only on the 1st of October, where there was a more "frontal" wind direction, where the disturbance from the scaffolding should be, in fact, less. Flag 2 fluxes were already excluded from analysis since, for microbial gradient method, they do correspond to moments where the two Burkard samplers are below the MRG and thus not providing reliable fluxes.

It is important to note that EC and FG footprints are, for stable stratification, the same when the EC measurements are made at the arithmetic mean of the highest and lowest gradient measurement heights. Conversely, for unstable stratification, the footprints match when the EC measurements are made at the geometric height of the aforementioned heights (Horst, 1999). Given the height differences between the EC and FG systems, the FG footprints should be smaller (Horst, 1999), receiving more contribution from the grassland itself and being less influenced by external influences and roughness elements.

---

## Author Response (AR2)

**Author's response to the Editor**

Co-Editor's comment is hereby reported in **boldfaced red** and it is followed by the authors' response along with eventual modification that were made to the paper.

**Thank you for your detailed responses to the referees. At this stage I am happy to accept it for publication in ACP. A few minor technical points remain. Please increase the font size of the new Figure 4, and make sure to upload figure files with much higher resolution in all cases.**

The authors wish to thank both the co-editor and the reviewers for their extremely useful comments, corrections and discussion. We do really believe that they made our paper better.

As for the specific comments of the co-editor, figure 4 has been update to have a larger font size (see track edited version of the manuscript) and all the figures where redrafted in high-resolution pdf format (see attached zip file).

While performing the aforementioned corrections the author's noticed minor spelling errors in the paper which have been corrected and the most relevant are:

- A reference (Weil et al., 2017) was missing at page 2 line 9 about long range transport of microorganisms and has been, therefore added.
- Italic formatting was missing on some of the equation terms in the text at page 6 and 7 and have been correctly italicized
- Some references reported abbreviated journal names which have been extended for consistency

[revised manuscript text omitted]